# Agentic Model Predictive Questioning Control in Visual Design

**Kuang-Da Wang** [* † 1 2] **Zhao Wang** [* ‡ 2] **Wei-Yao Wang** [2] **Yotaro Shimose** [2] **Jaechang Kim** [2] **Shingo Takamatsu** [2]

## Abstract

Recent Large Language Model based approaches for clarifying visual design largely focus on selecting questions that better uncover user intent, but often overlooks the *cognitive burden* imposed on users, i.e., the effort required to interpret and answer these questions, which is crucial for effective human-agent interaction. In this paper, we propose *Agentic Model Predictive Questioning Control (A-MPQC)*, a test-time framework that reduces proxy-estimated user interaction burden while improving visual design alignment by formulating multi-round clarification as trajectory optimization with receding-horizon replanning to revise its questioning strategy. In addition, we introduce lookahead *question plans* to reduce ambiguity early, and a lightweight respond-or-reject surrogate reward to steer questions toward lower user-burden formats (e.g., yes/no). Experiments on webpage and ad banner generation benchmarks show that *A-MPQC* not only generates designs better aligned with user intent, but also achieves lower user-interaction cost across diverse interaction baselines, including fixed-format strategies (e.g., multiple-choice and open-ended) and a retrieval-augmented baseline, without retraining. This paper sets a new perspective that explicitly *formulates* and *optimizes* the human cognitive burden jointly with final design alignment, opening new opportunities to advance human-agent interaction. Our code is publicly available at https://github.com/sony/a_mpqc

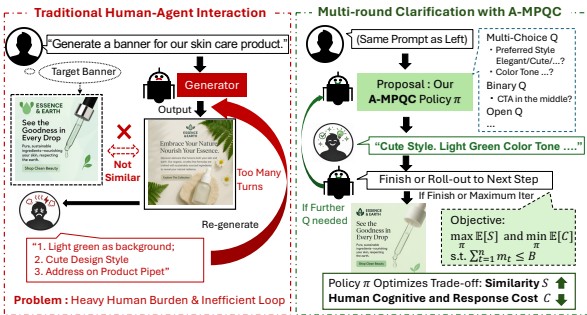

*Figure 1.* **Left:** Traditional human-agent interaction in visual design often requires many turns and heavy human burden. **Right:** *Our A-MPQC* iteratively updates a design plan via clarification questions under a fixed budget. It adapts questions at test time to improve alignment while reducing human cost.

## 1. Introduction

In visual design tasks (e.g., banner images and webpages), clients of designers (or users of generative models) often

---

[*]Equal contribution [†]Work done as a research intern at Sony Group Corporation. [1]National Yang Ming Chiao Tung University, Hsinchu, Taiwan [2]Sony Group Corporation, Tokyo, Japan. Correspondence to: Zhao Wang <Zhao.Wang@sony.com>.

*Proceedings of the 43rd International Conference on Machine Learning*, Seoul, South Korea. PMLR 306, 2026. Copyright 2026 by the author(s).

have a clear mental target in mind, but it is difficult to communicate these requirements precisely in a single prompt to a designer or an agent. As a result, prompt-based design frequently devolves into trial-and-error revision in practice (Hahn et al., 2025; Lu et al., 2026). One Example is illustrated in the left panel of Fig. 1, text-only prompting is inherently underspecified: many important requirements (e.g., tone, audience, visual style, emphasis, and other subjective preferences) remain implicit and may be inferred inconsistently by the model (Yang et al., 2025). This misalignment causes outputs to diverge from user intent and, in turn, require multiple refinement turns, imposing a substantial human burden.

Motivated by the misalignment issue, recent works study *learning-to-ask*: training agents to decide when clarification is needed and what to ask. Typically, these methods learn clarification policies from multi-turn supervision or preference signals (Zhang et al., 2025; Chen et al., 2025), or select questions using probabilistic or information-theoretic criteria. Active preference learning and RLHF aim to reduce human-feedback cost by strategically choosing which queries or comparisons to request (Muldrew et al., 2024; Ji et al., 2025).

Prior work suggests that clarification questions can help resolve underspecified user intent and improve interaction efficiency under uncertainty (Hahn et al., 2025; Wang et al., 2025b; Lei et al., 2020). Motivated by this perspective, a reasonable illustrative workflow in visual design is to elicit intent through a *progressive* questioning sequence: starting

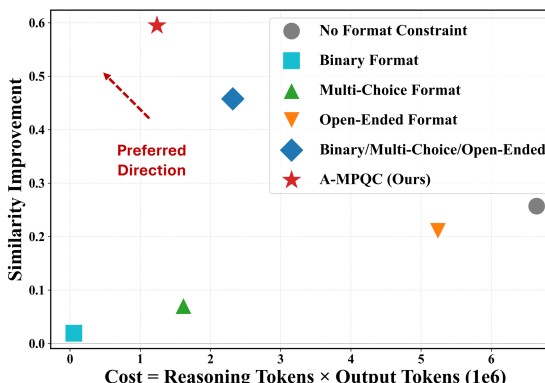

*Figure 2.* Cost-quality tradeoff of questioning strategies for banner generation. Each point shows the similarity improvement over a no-user-interaction baseline versus the user interaction cost.

with broad open-ended questions (e.g., key message and overall style), then moving to structured multiple choice (MC) questions to narrow high-level preferences (e.g., tone, palette, layout), and finally asking low-effort binary questions to finalize concrete details (e.g., CTA placement or headline position). This illustrative progression highlights the underlying cost-quality tradeoff: more expressive questions can reduce ambiguity and improve alignment, but they also increase user burden, whereas constrained formats lower response effort but may leave important intent unspecified. Fig. 2 summarizes this tradeoff across different question-asking strategies and motivates our proposed test-time optimization via different question formats.

These factors motivate our study from two perspectives: (i) while learning-to-ask and clarification methods are effective, many existing approaches rely on multi-turn supervision, preference signals, or task-specific interaction data for training or policy learning (Zhang et al., 2025; Chen et al., 2025; Muldrew et al., 2024). This makes them non-trivial to directly transfer to multimodal visual-design workflows, where collecting rich trajectories (e.g., intermediate images, edits, or preference feedback) is resource-intensive (Kirstain et al., 2023; Xu et al., 2023) and may require additional task-specific adaptation; and (ii) interactive generation also incurs a non-trivial *human-side cost*-users must spend time and cognitive effort to read, interpret, and answer clarification questions, yet this cognitive burden is rarely modeled or optimized in previous works.

Recent studies report a positive correlation between LLM reasoning effort (e.g., reasoning-token usage) and human cognitive effort (de Varda et al., 2025; Davidson, 2025), where model-side reasoning is associated with longer human reaction time and higher perceived demand. This provides a basis for studying and reducing human cognitive burden: we can use the agent's reasoning-token usage as a measurable proxy for the user's cognitive effort when designing burden-

aware questioning policies.

In this paper, we study multi-round clarification for visual design under a fixed *question budget*, i.e., the maximum number of user queries (distinct from cognitive/response cost). An agent iteratively asks clarification questions to reduce ambiguity, and a final artifact is generated once after the interaction, as illustrated in Fig. 1.

Within this setting, we propose *Agentic Model Predictive Questioning Control (A-MPQC)*, a test-time interaction framework that treats multi-round clarification as a trajectory optimization problem. *A-MPQC* adopts the receding-horizon template of Model Predictive Control (MPC) (Camacho & Bordons, 2007; Kouvaritakis & Cannon, 2015), which is widely used in control and RL/robotics to plan *action* rollouts using a (often training-learned) dynamics model. In contrast, *A-MPQC* rolls out *questions* to update intent beliefs and replans based on user responses. Beyond formulating clarification as trajectory optimization, *A-MPQC* is equipped with two test-time mechanisms: First, we introduce *question plans*, which pair each candidate question with a *remaining question plan* that states the intended follow-up direction, enabling ambiguity reduction with fewer interactions. Second, each query is posed in a respond-or-reject form, where the user may answer or decline; we extract a binary surrogate reward that indicates whether the direction is worth pursuing and use it to steer subsequent questions toward lower user-burden formats (e.g., yes/no and multiple-choice).

In summary, our contributions are threefold:

- **Problem Novelty: A cost-aware formulation for visual design questioning.** We introduce a multi-round clarification setting and evaluation framework for visual design generation that explicitly models the tradeoff between final design alignment and measured user interaction burden.

- **Method Novelty: A test-time questioning policy for visual design generation (A-MPQC).** We formulate multi-round clarification as trajectory optimization, and introduce three designs: (i) *question plans* that provide lookahead via a question with a remaining question plan; (ii) a respond-or-reject *surrogate reward* that guides replanning; and (iii) adaptive question formats that shift toward lower user-burden queries.

- **Experimental Novelty: Cross-modal evidence for format-aware questioning control.** Across webpage and ad banner generation benchmarks, *A-MPQC* consistently outperforms static questioning policies and fixed question formats, achieving higher design alignment under lower user burden via test-time format-aware control.

## 2. Related Work

**Human-in-the-Loop and Interactive LLM Agents.** Recent LLM-agent work integrates humans through feedback mechanisms such as preference supervision and clarification, with surveys summarizing this design space (Liu et al., 2025). Preference-based alignment studies data-efficient querying of human preferences (Muldrew et al., 2024) and online iterative RLHF that repeatedly updates policies via preference oracles (Ye et al., 2024). Complementary clarification-focused agents elicit latent intent by asking questions, including open-ended preference elicitation (Li et al., 2025a), training models to ask clarifying questions (Andukuri et al., 2024), deciding when to query under unclear instructions (Wang et al., 2025b), and interactive disambiguation for assistants (Murzaku et al., 2025). Finally, multi-turn evaluation benchmarks assess interactive tool use and feedback responsiveness beyond single-turn accuracy (Wang et al., 2024; Li et al., 2025c). In contrast to prior interactive pipelines that mainly optimize correctness/alignment or feedback efficiency—often requiring *task-specific training data*—we treat *human burden* as a constrained resource and propose a *test-time* questioning policy that jointly optimizes final alignment and user cost for intent-sensitive (and multimodal) design.

**Reasoning Tokens as Cognitive Effort Proxies.** We approximate user burden with model-side effort by using an LLM as a proxy for the user: reasoning-token usage estimates *thinking* cost and output length estimates *response* cost (Hahn et al., 2025). Evidence suggests that the cost of thinking scales similarly between large reasoning models and humans (de Varda et al., 2025), and that model reasoning effort predicts human decision time in a real-world decision task (Davidson, 2025). Together with a dual-process perspective on resource-limited reasoning (Gorelik, 2025), these results motivate token-level effort as a lightweight surrogate for human cognitive and response burden.

**Test-Time Optimization.** Test-time optimization improves model behavior without additional training by allocating extra computation during decoding. A common baseline is Best-of-$N$, which samples multiple candidates and selects the one with the highest reward. Beyond sampling, recent methods integrate reward models into decoding (Khanov et al., 2024; Kong et al., 2024; Xu et al., 2025) or search (Li et al., 2024; Qiu et al., 2025) for finer-grained control. Li et al. (2025b) takes a distinct approach, transforming reward feedback into textual gradient and using these to test-time update the response. Model Predictive Control (MPC) (Camacho & Bordons, 2007; Kouvaritakis & Cannon, 2015), originally developed in control domain, has also been applied to interactive settings with human feedback, where agents rely on learned dynamics or feedback models to anticipate future outcomes (Liang et al., 2024). Recent work further connects MPC to test-time optimization for language models via finite-horizon rollouts and receding-horizon replanning (Wang et al., 2026). However, these approaches typically require learned dynamics or reward models to evaluate rollouts, which is difficult to justify in our setting where collecting the interaction data needed to learn such models is expensive. In this work, we instead treat *clarification* as the object of test-time optimization: we optimize the sequence of questions asked to the user, enabling MPC-style replanning without training a dynamics or reward model.

## 3. Problem Setting

We study test-time human-agent interaction for visual design tasks, focusing on interaction patterns that clarify user intent *before* generation. We model an interaction loop that consists of (i) asking questions, (ii) receiving answers, and (iii) updating an explicit design plan. After $n$ rounds of plan updates, the system performs a single generation step based on the final plan. Importantly, we intentionally decouple *intent elicitation* from downstream artifact refinement: questions update only the plan, and generation is performed once using the final plan. We adopt this decoupling to isolate the effect of questioning policies on intent elicitation, avoiding confounding gains from iterative refinement.

### 3.1. Question Agent and User Agent

We introduce two agents to separate *what the user wants* from *what the system can infer*: the user agent represents a user with a intent of target design, while the question agent is the policy that elicits intent and maintains an explicit design plan. The user agent has access to a target design $x^\star$, instantiated as a reference image and an underlying detailed requirement specification. The question agent never observes $x^\star$ and must infer the requirements by asking questions and updating a design plan $d$. Interaction starts from an initial underspecified prompt $p_0$ provided by the user agent. Based on $p_0$, the question agent constructs an initial design plan $d_0$. The design plan serves as an interpretable intermediate representation that bridges user intent and the final generated artifact.

### 3.2. Interaction Loop

We consider a fixed-budget interaction protocol with $n$ plan updates. At each round $t \in \{1, \ldots, n\}$, the number of questions $m_t$ is *adaptively* determined by the question agent based on the current plan state and accumulated interaction history, with $0 \le m_t \le m$. We index the questions within a round by $j \in \{1, \ldots, m_t\}$, so that at most $m$ questions can be asked per round. Within each round, questions are asked sequentially and the agent may early stop, so $m_t$ is the realized number of executed queries. Let $q_{t,j}$ be the $j$-th question at round $t$, and let $a_{t,j}$ be the user response. After

*Figure 3.* In standard MPC (e.g., robot teaching with human feedback), an agent uses a learned dynamics model to roll out candidate trajectories, executes the first action of the best rollout, and repeats in a receding-horizon manner. *A-MPQC* extends this principle to multi-round, test-time clarification by rolling out *question plans* $\tilde{q} = (q, \rho)$ that summarize questioning directions to reduce ambiguity. A binary `respond`/`reject` surrogate reward indicates whether a direction is worth pursuing, helping avoid costly or unproductive queries. Accepted questions update the design plan, while rejected directions trigger replanning over question content and format.

the within-round $m_t$ questions finish, the question agent updates the plan:

$$d_t = U\big(d_{t-1}, \{(q_{t,j}, a_{t,j})\}_{j=1}^{m_t}\big),$$

where $U(\cdot)$ is a plan update procedure implemented by a frozen LLM conditioned on the current dialogue context.

After $n$ rounds, the question agent outputs a final plan $d_n$. A LLM generator $G$ produces the final artifact:

$$x = G(d_n).$$

In this paper, we do not fine-tune $U$ or $G$. Due to data and cost constraints, we focus on inference-time optimization of interaction patterns that collect information efficiently.

### 3.3. Human Interaction Cost

Human involvement incurs two types of cost: (i) *cognitive cost* to interpret and reason about a question, and (ii) *response cost* to compose an answer. We operationalize both using token-based proxies. For each question–answer interaction $(t, j)$, let $T_{t,j}^{\text{think}}$ denote the reasoning tokens and $T_{t,j}^{\text{resp}}$ denote the answer response tokens. We define the interaction cost as:

$$C(q_{t,j}) = T_{t,j}^{\text{think}} \cdot T_{t,j}^{\text{resp}}, \quad C_{\text{total}} = \sum_{t=1}^{n} \sum_{j=1}^{m_t} C(q_{t,j}). \quad (1)$$

We use the product to highlight interactions that are both cognitively demanding and costly to answer.

**Why we do not charge question-input length.** One may consider adding an "input" token $T_{t,j}^{\text{in}}$ to model reading cost. We omit it for two reasons: (i) it largely overlaps with our cognitive-effort proxy $T_{t,j}^{\text{think}}$ and may double-count the same

underlying burden; and (ii) input length is a weak proxy for difficulty (a long question can still be easy), whereas reasoning effort better reflects human burden and has been empirically linked to human reaction time and cognitive demand (de Varda et al., 2025; Davidson, 2025).

### 3.4. Performance Metric and Objective

Performance is measured by similarity between the generated artifact $x$ and the user target $x^\star$:

$$S(x, x^\star) = \text{Sim}(x, x^\star),$$

where Sim is computed by a vision-language model (see Sec. 5 for details). This similarity is used for evaluation and is not revealed to the question agent during interaction.

Let $B$ denote the *question budget*, i.e., the maximum number of questions allowed. Given this budget, our goal is to study the cost-performance tradeoff by designing a policy $\pi$ that maximizes alignment while minimizing proxy-estimated human interaction cost:

$$\max_{\pi} \mathbb{E}\big[S(x, x^\star)\big] \quad \text{and} \quad \min_{\pi} \mathbb{E}\big[C_{\text{total}}\big] \quad \text{s.t.} \quad \sum_{t=1}^{n} m_t \leq B.$$

## 4. Method

This section presents **Agentic Model Predictive Questioning Control (*A-MPQC*)**, an MPC-inspired test-time interaction method for cost-efficient intent elicitation in visual design. We first review trajectory optimization and Model Predictive Control (MPC) (Sec. 4.1). We then present *A-MPQC* as an MPC-inspired formulation and discuss its differences from standard MPC (Sec. 4.2). Next, we introduce three designs that make MPC-style planning feasible for multi-round clarification (Secs. 4.2.1 to 4.2.3). Finally, we summarize the algorithm (Sec. 4.3).

## 4.1. Preliminary: Trajectory Optimization and Model Predictive Control

Trajectory optimization seeks an action sequence that maximizes cumulative reward. Given state $s_t \in \mathcal{S}$, action $u_t \in \mathcal{U}$, and per-step reward $r$, an objective over horizon $T$ is

$$u^\star_{0:T-1} \in \arg \max_{u_{0:T-1}} \sum_{t=0}^{T-1} r(s_t, u_t). \qquad (2)$$

MPC approximates long-horizon optimization by repeatedly solving a *finite-horizon* problem, executing the first action, and replanning from the updated state. At time $t$, MPC solves

$$u^\star_{t:t+H-1} \in \arg \max_{u_{t:t+H-1}} \sum_{h=0}^{H-1} r(s_{t+h}, u_{t+h}), \qquad (3)$$

executes $u^\star_t$, and repeats. Figure 3 illustrates the receding-horizon MPC loop and how we transpose it to multi-round clarification, in contrast to prior MPC interactive control (e.g., robot teaching with human feedback) that uses a learned interaction dynamics model to roll out future feedback under candidate actions.

## 4.2. Agentic Model Predictive Questioning Control

*A-MPQC* adopts an interaction-level MPC-style replanning loop for costly human-in-the-loop clarification, as illustrated in Fig. 3. At each round, the current design plan serves as the state, a proposed questioning direction serves as the action, and user feedback provides a costly surrogate reward for deciding whether that direction should update the plan. The loop consists of: *(i) plan* by proposing question plans that pair the next question with a compact lookahead over the remaining questioning direction; *(ii) evaluate* the proposed direction through respond-or-reject user feedback; *(iii) execute* accepted questions by using their QA pairs to update the design plan; and *(iv) replan* the next questioning strategy by conditioning on the updated plan and all previous accept/reject outcomes.

Unlike standard MPC, which relies on cheap simulators for evaluation, A-MPQC minimizes real-world interaction costs through questions plans and surrogate feedback. This enables receding-horizon adaptation without requiring user evaluations of whole trajectories.

**A-MPQC Formulation and MPC-Inspired Correspondence.** We extend our multi-round clarification setting from the standard MPC formulation as follows. At round $t$, the *state* corresponds to the current design plan $d_t$, which summarizes the agent's belief about user intent. An *action* corresponds to proposing a question $q_{t,j}$. Executing an action corresponds to a single state transition: posing $q_{t,j}$ elicits a user response $a_{t,j}$, and the state is updated

as $d_{t+1} = U(d_t, \{(q_{t,j}, a_{t,j})\})$. We use $m_t$ to denote the number of questions at round $t$.

**Three Designs That Enable MPC-Style Questioning.** In standard MPC, candidate trajectories can be easily evaluated by simulating a dynamics model. In interactive questioning, however, evaluating trajectories would require extra user cost. To make MPC-style planning more efficient, *A-MPQC* introduces the following three design choices.

### 4.2.1. DESIGN I: INTERACTION-LEVEL AND LOW-COST MPC VIA QUESTION PLANS

At round $t$, *A-MPQC* queries the user *sequentially* up to a per-round cap $m$ and may terminate early, yielding a realized number of queries $m_t \le m$. At each within-round step $j \in [m_t]$, the agent proposes a *question plan*

$$\tilde{q}_{t,j} \triangleq (q_{t,j}, \rho_{t,j}), \qquad j = 1, \ldots, m_t, \qquad (4)$$

where $q_{t,j}$ is the next question and $\rho_{t,j}$ is a *remaining question plan* that specifies the intended follow-up direction (i.e., a short-horizon lookahead). Concretely, $\tilde{q}_{t,j}$ is generated conditioned on the current plan and the within-round history (via the current questioning strategy $\pi_t$), enabling the agent to reason about downstream effects *before* committing to the next query, without requiring additional user interactions.

### 4.2.2. DESIGN II: BINARY SURROGATE REWARD VIA RESPONSE-OR-REJECT FORMAT

To evaluate a question plan without requiring full answers for all candidate directions, each $\tilde{q}_{t,j}$ is posed in a *respond-or-reject* form. This format elicits a *single* user feedback: the user either answers the proposed question or declines to answer, so each interaction still corresponds to one question. We denote the feedback by $a_{t,j} \in \mathcal{A}$ and use it in two ways: (i) we extract a binary surrogate reward that indicates whether the proposed direction is worth pursuing, and (ii) when accepted, we reuse $a_{t,j}$ as the substantive answer for updating the design plan.

Formally, let $g : \mathcal{A} \to \mathcal{R}$ be a deterministic parser that maps the response to a binary outcome,

$$r_{t,j} \triangleq g(a_{t,j}) \in \mathcal{R}, \quad \mathcal{R} \triangleq \{\texttt{respond}, \texttt{reject}\}. \quad (5)$$

We execute only accepted queries and define the accepted index set:

$$\mathcal{A}_t \triangleq \{ j \in [m_t] \mid r_{t,j} = \texttt{respond} \}. \qquad (6)$$

The accepted QA pairs $\{(q_{t,j}, a_{t,j})\}_{j \in \mathcal{A}_t}$ are used to update the design plan, inducing a state transition from $d_t$ to $d_{t+1}$. Maximizing the $\texttt{respond}$ rate biases the policy toward directions the user is willing to answer. Importantly, respond-or-reject is not free: *every* feedback $a_{t,j}$ (including $\texttt{reject}$) is treated as a user interaction and incurs both cognitive and response cost under our token proxies.

---

**Algorithm 1** Agentic Model Predictive Questioning Control

**Input:** Initial prompt $p_0$, plan updater $U$, generator $G$, rounds $n$, per-round cap $m$

**Output:** Generated artifact $x$

1  Initialize $d_0 \leftarrow U(\varnothing, p_0)$  Initialize $\pi_0$
2  **for** $t = 0$ **to** $n - 1$ **do**
3  $\quad$ $j \leftarrow 1$ **while** $j \leq m$ **do**
4  $\quad\quad$ **if** $StopCriterion\big(d_t, \{(q_{t,k}, a_{t,k})\}_{k=1}^{j-1}\big)$ **then**
5  $\quad\quad\quad$ **break**
6  $\quad\quad$ $\tilde{q}_{t,j} = (q_{t,j}, \rho_{t,j}) \sim \pi_t(\cdot \mid d_t, \{(q_{t,k}, a_{t,k})\}_{k<j})$
7  $\quad\quad$ Obtain user feedback $a_{t,j}$
8  $\quad\quad$ $r_{t,j} \leftarrow g(a_{t,j})$ $\quad$ // surrogate reward
9  $\quad\quad$ $j \leftarrow j + 1$
10  $\quad$ $m_t \leftarrow j - 1$
11  $\quad$ $\mathcal{A}_t \leftarrow \{\, j \in [m_t] \mid r_{t,j} = \texttt{respond}\,\}$
12  $\quad$ Update $d_{t+1} \leftarrow U\Big(d_t, \{(q_{t,j}, a_{t,j})\}_{j \in \mathcal{A}_t}\Big)$
13  $\quad$ $\pi_{t+1} \leftarrow \Phi\big(d_{t+1}, \pi_t, \tilde{q}_{t,1:m_t}, a_{t,1:m_t}\big)$
14  $x \leftarrow G(d_n)$

---

#### 4.2.3. DESIGN III: ADAPTIVE REPLANNING OVER QUESTIONING CONTENT AND FORMAT

After updating $d_{t+1}$, *A-MPQC* replans by updating its questioning strategy for the next round. Specifically, it jointly conditions on the updated plan and the entire within-round interaction history, i.e., $(d_{t+1}, \tilde{q}_{t,1:m_t}, a_{t,1:m_t})$, to produce a revised strategy:

$$\pi_{t+1} \triangleq \Phi\big(d_{t+1}, \pi_t, \tilde{q}_{t,1:m_t}, a_{t,1:m_t}\big). \tag{7}$$

Here, $\Phi$ is a test-time replanning operator that updates the policy $\pi$. Intuitively, $\texttt{respond}$ (i.e., $g(a_{t,j}) = \texttt{respond}$) outcomes reinforce useful directions under $d_{t+1}$, while $\texttt{reject}$ outcomes indicate that a direction or its current question is not worth pursuing, prompting the next-round strategy $\pi_{t+1}$ to revise accordingly.

We consider three question formats: Binary (i.e., yes/no question), Multiple choice, and Open-Ended (free-form response). We treat binary questions as the least expressive, multiple-choice as intermediate, and open-ended as the most expressive. During replanning, the agent is instructed to use the least expressive format sufficient for the ambiguity, and to increase expressiveness only if a less expressive format is rejected. This promotes convergent clarification while keeping constrained queries as the default.

### 4.3. Algorithm and Cost Efficiency

Algorithm 1 summarizes *A-MPQC*. At each round $t$, the agent runs a *sequential* within-round loop: it proposes a question plan, queries the user, and receives a single textual feedback $a_{t,j}$. From this feedback, the agent extracts a

binary surrogate reward $r_{t,j} = g(a_{t,j})$; if $\texttt{respond}$, the same $a_{t,j}$ provides the substantive answer used for plan update. Within a round, the agent may early stop based on the current plan and accumulated dialogue, yielding an adaptive number of queries $m_t$ with $0 \leq m_t \leq m$. After the within-round sequence terminates, the plan is updated from $d_t$ to $d_{t+1}$ using accepted QA pairs $\{(q_{t,j}, a_{t,j})\}_{j \in \mathcal{A}_t}$, where $\mathcal{A}_t \triangleq \{ j \in [m_t] \mid g(a_{t,j}) = \texttt{respond}\}$. Finally, replanning updates the next-round questioning strategy via $\pi_{t+1} = \Phi(d_{t+1}, \pi_t, \tilde{q}_{t,1:m_t}, a_{t,1:m_t})$.

**Why not directly apply MPC on the within-round $m$-question dimension.** Unlike standard MPC, where rollouts can be simulated cheaply using a dynamics model, here each rollout step requires additional human interaction. If we apply MPC to a length-$H$ within-round question sequence and evaluate $K$ candidates, the cost scales with the number of rollouts: in the worst case, up to $\mathcal{O}(KH)$ user responses per step. Even if we replace explicit rollouts with our proposed *question plans*, evaluating $K$ candidates still incurs $\mathcal{O}(K)$ extra user responses at every step, which can be substantial in practice. *A-MPQC* avoids this bottleneck by applying MPC to the *interaction process* rather than to within-round question sequences. Specifically, we treat the interaction as a length-$n$ trajectory (one plan update per round), use $m$ candidates per round, and replace horizon-$H$ rollouts with compact *question plans* that summarize short-horizon trajectories. Therefore, the total number of user responses remains bounded by the question budget $B$.

## 5. Experiments

### 5.1. Experimental Setup

#### 5.1.1. DATASETS

We evaluate on two visual design tasks across modalities: **WebGen-V (Wang et al., 2025a) (webpage HTML generation).** Each instance specifies a target webpage image $x^\star$; the agent elicits requirements and generates an HTML webpage to match $x^\star$.

**MIMO (Wang et al., 2025c) (banner image generation).** Each instance specifies a target banner image $x^\star$; the agent elicits user intent and generates an image aligned with $x^\star$.

We measure alignment by similarity $S(x, x^\star)$ using an LLM judge that rates the generated artifact $x$ against the target $x^\star$ on five aspects (1-5 each) and averages across aspects,

$$S(x, x^\star) = \frac{1}{5} \sum_{k=1}^{5} s_k(x, x^\star), \quad s_k \in \{1, 2, 3, 4, 5\}.$$

We report improvement over the *no user interaction* baseline $x_{\text{no-int}}$ as $\Delta S \triangleq S(x, x^\star) - S(x_{\text{no-int}}, x^\star)$, and efficiency $\Delta S / \tilde{C}$, where $\tilde{C}$ is the normalized interaction cost computed from Eq. (1) with $\tilde{C} \triangleq C_{\text{total}} / 10^6$.

### 5.1.2. BASELINES

All methods use the same frozen plan updater $U$ and generator $G$, and are evaluated under the same budget $B{=}12$[1]. The corresponding prompt templates and format constraints are detailed in Appendix B.1.

**No user interaction.** Single-shot generation from the initial plan $d_0$ without any interaction.

**Direct generation (DG).** A baseline *does not* enforce a fixed question format and generates questions based on the question-asking principles (Hahn et al., 2025).

**DG variants.** We use **DG** as a base interactive baseline and instantiate five variants by controlling its response format and retrieval augmentation.

- **DG + Binary:** Fixed yes/no format.
- **DG + Multiple-Choice:** Fixed multiple-choice format.
- **DG + Open-Ended:** Fixed free-form text format.
- **DG + Flexible:** Selects among binary, multiple-choice, and open-ended formats.
- **DG + Flexible + RAG:** DG + Flexible augmented with an external interaction QAs for in-context learning.

### 5.1.3. IMPLEMENTATION DETAILS

**Agent Backbones.** The user agent uses Gemini-2.5-Pro (Gemini Team, 2025) across all experiments to provide consistent deliberation token signals for cost estimation. The question agent backbone is varied to test robustness; the main paper reports results with Gemini-2.5-Pro, and results with GPT-5.2 (OpenAI, 2025) are deferred to Appendix A.

**Generation and Evaluation.** We follow the original benchmark protocols for generation and judging. For WebGen-V, we use Gemini-2.5-Pro for both final HTML generation and LLM judging, following the WebGen-V evaluation protocol. For MIMO, we use gpt-image-1 for final image generation and GPT-4.1 as the judge, following the MIMO protocol. The evaluation and generation prompts are detailed in Appendix B.3 and Appendix B, respectively.

**Retrieval baseline (DG + Flexible + RAG).** We construct a question pool by running DG+Binary, DG+Multiple-Choice, and DG+Open-Ended and storing their interactions. We embed each trace with all-MiniLM-L6-v2 (Reimers & Gurevych, 2019) and retrieve top-3 traces using the embedding of the current context (design plan and QA pairs). Retrieved traces are included as in-context examples for DG+Flexible.

---

[1] Hahn et al. (2025) run 15-turn conversations for evaluation and report that key alignment signals often plateau after 10 interactions. We set $B{=}12$ as a conservative middle ground.

### 5.2. RQ1: Does *A-MPQC* yield better tradeoffs?

Table 1 compares *A-MPQC* with baselines under the same question budget. We allocate the budget as $B = n \cdot m$ with $n{=}3$ replanning rounds and $m{=}4$ questions per round. Across both design tasks, *A-MPQC* attains strong similarity improvements while maintaining substantially lower normalized interaction cost than unconstrained direct questioning. In particular, *A-MPQC* achieves the best efficiency $\Delta S/\tilde{C}$, indicating a more favorable alignment-burden tradeoff. We observe that the absolute interaction cost is generally higher on WebGen-V than on MIMO. This is expected because HTML webpage generation involves richer and more coupled constraints (e.g., multi-section structure, layout hierarchy, typography, and asset usage), which typically require more detailed reasoning from the user.

The baselines further contextualize these gains. First, DG behaves more like an unconstrained conversation: without explicit format control, it often relies on conversation to elicit intent, which increases user burden even when the remaining uncertainty could be resolved with constrained queries. Second, fixed-format strategies exhibit a clear cost-quality tradeoff: binary questions are low-cost but often insufficient, whereas open-ended questions can capture richer intent at substantially higher user burden. Allowing format flexibility (DG+Flexible) partially mitigates this by selecting formats based on remaining uncertainty, but it lacks an explicit mechanism to filter out or quickly redirect unproductive questioning directions. Finally, retrieval augmentation (DG+Flexible+RAG) can improve performance in some cases, but its effect is inconsistent when retrieved traces do not match the current design intent. We include qualitative examples in Appendix C.

### 5.3. RQ2: How should a budget be allocated?

We fix the question budget to $B{=}12$ and vary how it is split into the number of plan-update rounds $n$ and the per-round cap $m$. We evaluate $(n, m) \in \{(12, 1), (6, 2), (3, 4), (2, 6), (1, 12)\}$ and visualize the results in Fig. 4. The best performance comes from intermediate allocations for *A-MPQC*.

When $n$ is too small (e.g., $1{\times}12$), the agent asks many questions without updating the plan in between, so later questions are based on an outdated understanding and cannot correct early mistakes. When $n$ is too large (e.g., $12{\times}1$), each plan update is made after only one question, which provides too little evidence and makes the updates noisy. As a result, allocations like $3{\times}4$ perform better because they balance (i) asking a few related questions before updating and (ii) updating the plan multiple times to adapt the next questions. *A-MPQC* peaks at $3{\times}4$ in similarity improvement (left), while $2{\times}6$ often offers the best efficiency (right) because it preserves most of the improvement but with lower

*Table 1.* Comparison of cost-performance trade-offs across different methods under a fixed budget. $\tilde{C} = C_{\text{total}}/10^6$ and $\Delta S$ is improvement over *no user interaction*. The best and second-best results are highlighted in **bold** and underlined, respectively.

| WebGen-V (Web HTML Generation) | | | | MIMO (Banner Image Generation) | | | |
|---|---|---|---|---|---|---|---|
| Method | $\tilde{C}\downarrow$ | $\Delta S\uparrow$ | $\Delta S/\tilde{C}\uparrow$ | Method | $\tilde{C}\downarrow$ | $\Delta S\uparrow$ | $\Delta S/\tilde{C}\uparrow$ |
| No user interaction | 0.000 | 0.000 | - | No user interaction | 0.000 | 0.000 | - |
| DG (Direct Generation) | 6.046 | 0.366 | 0.060 | DG (Direct Generation) | 6.646 | 0.256 | 0.038 |
| DG + Binary | **0.116** | 0.017 | 0.150 | DG + Binary | **0.055** | 0.019 | 0.342 |
| DG + Multiple-Choice | 6.449 | 0.295 | 0.045 | DG + Multiple-Choice | 1.614 | 0.070 | 0.043 |
| DG + Open-Ended | 13.276 | **0.566** | 0.042 | DG + Open-Ended | 5.240 | 0.209 | 0.040 |
| DG + Flexible | 9.603 | 0.418 | 0.043 | DG + Flexible | 2.318 | 0.457 | 0.197 |
| DG + Flexible + RAG | 7.298 | 0.383 | 0.052 | DG + Flexible + RAG | 1.331 | 0.292 | 0.219 |
| A-MPQC (Ours) | 2.548 | 0.480 | **0.188** | A-MPQC (Ours) | 1.238 | **0.594** | **0.480** |

cost (middle). For DG and DG+Open-Ended, we observe that the interaction cost can decrease as updates become less frequent (i.e, moving toward $1\times12$), because these policies often satisfy earlier and therefore do not always exhaust the full question budget.

### 5.4. RQ3: Which components drive *A-MPQC*'s gains?

Table 2 ablates the three designs of *A-MPQC* on MIMO under fixed $B=12$ with $(n,m)=(3,4)$ (cf. RQ1). We also report two diagnostic variants. *A-MPQC*-on-$m$ applies MPC-style control over the within-round question dimension, which changes the control dimension rather than removing a design component. *A-MPQC*-IgnoreReject keeps the same respond-or-reject user policy, but prevents the agent from using rejection feedback for replanning. The full *A-MPQC* attains the strongest alignment gain ($\Delta S$) and best efficiency ($\Delta S/\tilde{C}$) at moderate user interaction cost ($\tilde{C}$).

The ablations show the contribution of each design:

- **_A-MPQC_-on-$m$.** Applying MPC-style control over the within-round question dimension achieves a relatively high $\Delta S$, but it substantially increases interaction cost, resulting in lower efficiency. This indicates that within-round control is interaction-inefficient in our setting.
- **w/o Design I (no question plans).** Removing lookahead question plans reduces user burden, since the user no longer evaluates planned follow-up directions. However, it also decreases alignment, suggesting that question plans help the agent elicit more useful intent information.
- **w/o Design II (all-respond).** Replacing respond-or-reject feedback with always-answer behavior yields the lowest cost, but its alignment gain drops substantially. This suggests that rejection feedback helps the agent avoid unproductive questioning directions.
- **_A-MPQC_-IgnoreReject.** This variant keeps the same respond-or-reject interaction protocol but ignores rejection outcomes during replanning. Its weaker alignment and efficiency compared with the full method indicate that rejection is useful not merely as a low-cost response

*Table 2.* Ablation study on MIMO under fixed budget $B=12$ with $(n,m)=(3,4)$. *A-MPQC*-on-$m$ changes the control dimension and is not a design removal. *A-MPQC*-IgnoreReject keeps the respond-or-reject user policy but does not use rejection feedback for replanning.

| Variant | $\tilde{C}\downarrow$ | $\Delta S\uparrow$ | $\Delta S/\tilde{C}\uparrow$ |
|---|---|---|---|
| *A-MPQC* (full) | 1.238 | **0.594** | **0.480** |
| *A-MPQC*-on-$m$ | 6.696 | 0.537 | 0.080 |
| w/o Design I (no question plans) | 0.925 | 0.391 | 0.422 |
| w/o Design II (all-respond) | **0.474** | 0.204 | 0.431 |
| *A-MPQC*-IgnoreReject | 0.861 | 0.311 | 0.361 |
| w/o Design III (direct generation) | 1.758 | 0.273 | 0.155 |

option, but as a control signal for updating subsequent questioning directions and formats.
- **w/o Design III (direct generation).** Removing adaptive format control yields a weaker tradeoff, highlighting the benefit of selecting the least-burdensome format that is sufficient for the current uncertainty.

## 6. Discussion

**Human evaluation of the interaction-cost proxy.** Our experiments use a token-based proxy to measure interaction cost, which enables controlled and reproducible comparisons across methods. To validate whether this proxy reflects perceived user burden, we conduct a human evaluation on sampled multi-turn interactions. Six domain-relevant expert evaluators with experience in agentic banner or webpage generation rate the burden of question–answer traces. We then compare the human ratings with the corresponding token-based signals using rank correlations.

As shown in Table 3, human-rated burden is positively correlated with the proposed proxy signals. Output tokens show a strong correlation with perceived response burden, while reasoning tokens show a moderate positive correlation with perceived reasoning burden. The combined proxy also correlates positively with overall perceived burden. These results support using the token-based cost as a practical

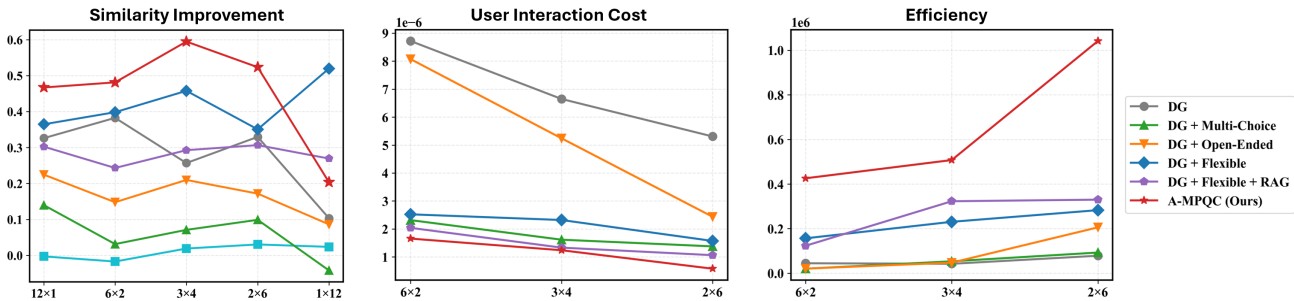

*Figure 4.* Effect of budget allocation under a fixed question budget $B{=}12$ with $B{=}n \cdot m$. **Left:** similarity improvement $\Delta S$ across allocations $(12, 1)$, $(6, 2)$, $(3, 4)$, $(2, 6)$, and $(1, 12)$. **Middle:** user interaction cost $\tilde{C}{=}C_{\text{total}}/10^6$. **Right:** efficiency $\Delta S/\tilde{C}$. We focus on non-extreme allocations and omit Binary-only when reporting cost/efficiency, as it can produce near-zero or negative $\Delta S$.

*Table 3.* Small-scale human validation of the interaction-cost proxy. We report rank correlations between human-rated burden and token-based proxy signals.

| Dimension | Proxy | Spearman $\rho$ | Kendall $\tau$ |
|---|---|---|---|
| Response | Output tokens | 0.71 | 0.49 |
| Reasoning | Reasoning tokens | 0.41 | 0.28 |
| Overall | $\log(1 + C)$ | 0.65 | 0.48 |

relative measure of interaction burden in our evaluation. Accordingly, our claims focus on reducing proxy-estimated interaction cost and improving alignment–cost tradeoffs, with human evaluation providing additional evidence that the proxy tracks perceived burden trends.

**Refusal-enabled baselines.** One may wonder whether the gains of *A-MPQC* mainly come from giving users a refusal option, which could mechanically reduce measured interaction cost. However, in *A-MPQC*, respond-or-reject feedback is not only a low-cost response channel, but also a surrogate control signal: rejection tells the agent that the current questioning strategy should be revised.

To separate these effects, we evaluate refusal-enabled DG baselines under the same refusal channel and cost accounting, with results reported in Appendix A.3. Refusal reduces the cost of several baselines and improves some flexible strategies, but *A-MPQC* still achieves the best alignment improvement and efficiency. Moreover, *A-MPQC*-IgnoreReject in Sec. 5.4 keeps the same respond-or-reject protocol but disables rejection-based replanning, and performs worse than the full method. These results suggest that the main benefit comes from using refusal feedback to optimize questioning direction and format, rather than from the refusal option alone.

**Limitations and future work.** This work focuses on visual design tasks where user intent can be represented by an explicit intermediate design plan. Such a plan is natural for webpage and banner generation, but other interactive generation or multimodal reasoning tasks, such as exploratory rea-

soning, open-ended creative collaboration, or fine-grained perceptual judgment, may not admit a single evolving plan state. Extending cost-aware questioning control beyond explicit design-plan representations is an important direction for future work.

Our protocol also separates intent elicitation from artifact refinement: the agent clarifies and updates the plan first, then generates the final artifact once. This isolates the effect of questioning policies, but real design workflows often involve iterative inspection, critique, and revision of intermediate outputs. Future work can combine *A-MPQC* with iterative visual refinement while still accounting for user-side burden.

Finally, although our human validation provides preliminary evidence that the token-based proxy tracks perceived burden, broader studies with professional designers and end users are needed to evaluate subjective satisfaction, response time, design quality, and adaptation to different user expertise levels, design domains, and interaction preferences.

## 7. Conclusion

We studied budgeted multi-round clarification for visual design generation, where an agent asks questions to update an explicit design plan and generates the final artifact once from the resulting plan while accounting for user-side burden. To optimize this process at test time without retraining, we proposed *Agentic Model Predictive Questioning Control (A-MPQC)*, which formulates clarification as trajectory optimization with receding-horizon replanning under a fixed question budget. *A-MPQC* realizes this idea with (i) question plans for lightweight lookahead, (ii) a respond-or-reject surrogate reward to steer questioning directions, and (iii) adaptive question formats to reduce user burden. Across webpage HTML generation and banner image generation, *A-MPQC* achieves better cost-quality tradeoffs than diverse questioning strategies and a retrieval-based baseline under matched budgets. We view *A-MPQC* as a step toward better human-agent co-creation, where interactions are optimized not only for output quality but also for the user experience.

## Impact Statement

This paper studies test-time optimization of multi-round clarification for human-agent visual co-creation, aiming to improve intent alignment while reducing interaction burden. Our experiments use large language models as proxy users, offering a useful but imperfect approximation; observed gains in alignment efficiency should not be treated as a substitute for real-user studies. If deployed responsibly, such methods could make design tools more usable and accessible, especially for non-experts. Potential risks include faster creation of persuasive or misleading visuals and amplification of biases from underlying models.

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

*Table 4.* Comparison of cost-performance trade-offs across different methods under a fixed budget, using GPT-5.2 as the question-agent backbone. $\tilde{C} = C_{\text{total}}/10^6$ and $\Delta S$ is improvement over *no user interaction*. The best and second-best results are highlighted in **bold** and underlined, respectively.

| WebGen-V (Web HTML Generation) | | | | MIMO (Banner Image Generation) | | | |
|---|---|---|---|---|---|---|---|
| Method | $\tilde{C} \downarrow$ | $\Delta S \uparrow$ | $\Delta S/\tilde{C} \uparrow$ | Method | $\tilde{C} \downarrow$ | $\Delta S \uparrow$ | $\Delta S/\tilde{C} \uparrow$ |
| No user interaction | 0.000 | 0.000 | - | No user interaction | 0.000 | 0.000 | - |
| DG (Direct Generation) | 11.696 | 0.203 | 0.017 | DG (Direct Generation) | 6.598 | 0.451 | 0.068 |
| DG + Binary | **0.108** | -0.275 | -2.543 | DG + Binary | **0.066** | -0.007 | -0.110 |
| DG + Multiple-Choice | 8.788 | 0.110 | 0.012 | DG + Multiple-Choice | 1.193 | 0.075 | 0.063 |
| DG + Open-Ended | 15.477 | 0.063 | 0.004 | DG + Open-Ended | 5.567 | 0.228 | 0.041 |
| DG + Flexible | 13.774 | 0.201 | 0.014 | DG + Flexible | 2.818 | 0.455 | 0.161 |
| DG + Flexible + RAG | 12.932 | 0.101 | 0.007 | DG + Flexible + RAG | 2.728 | 0.376 | 0.137 |
| *A-MPQC* (Ours) | 2.864 | **0.212** | **0.074** | *A-MPQC* (Ours) | 1.717 | **0.582** | **0.338** |

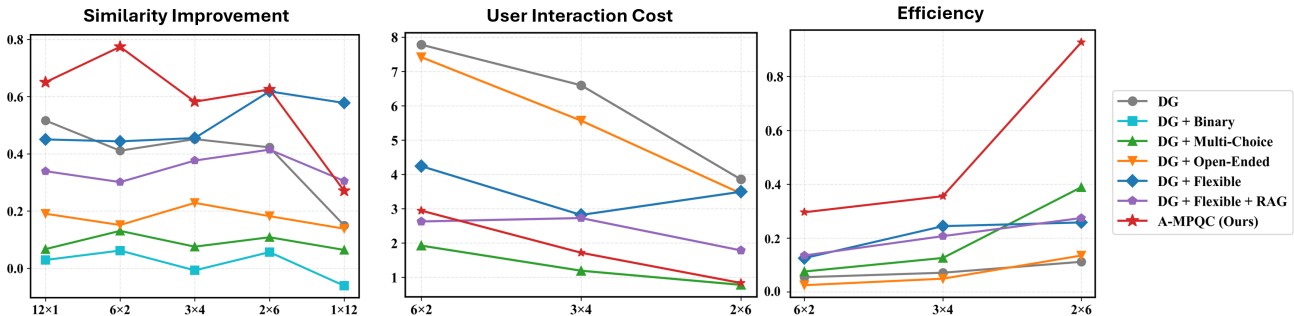

*Figure 5.* Effect of budget allocation under a fixed question budget $B=12$ with $B=n \cdot m$. **Left:** similarity improvement $\Delta S$ across allocations $(12, 1)$, $(6, 2)$, $(3, 4)$, $(2, 6)$, and $(1, 12)$. **Middle:** user interaction cost $\tilde{C}=C_{\text{total}}/10^6$. **Right:** efficiency $\Delta S/\tilde{C}$. We focus on non-extreme allocations and omit Binary-only when reporting cost/efficiency, as it can produce near-zero or negative $\Delta S$.

## A. Additional Experimental Results

This appendix reports additional results omitted from the main paper due to space. We (i) evaluate robustness of the questioning policy to the question-agent backbone by replacing Gemini-2.5-Pro with GPT-5.2 (Appx. A.1); and (ii) provide sensitivity analysis over how the fixed question budget is allocated across plan-update rounds and within-round caps (Appx. A.2). All other experimental components and protocols follow Sec. 5 (e.g., the user agent remains fixed for consistent cost-proxy measurement, and generation/judging follow the benchmark protocols for WebGen-V and MIMO).

### A.1. Results with GPT-5.2 as Question Agent Backbone

Table 4 reports the cost-performance tradeoff under the same executed-interaction budget $B=12$. On **MIMO**, *A-MPQC* maintains the strongest efficiency $\Delta S/\tilde{C}$, substantially outperforming both unconstrained direct questioning and fixed-format baselines. We use the same budget allocation as in RQ1, with $B = n \cdot m$ set to $n=3$ replanning rounds and $m=4$ questions per round.

### A.2. Sensitivity across Budget Allocation

Fig. 5 shows that the overall trend across allocations $B=n \cdot m$ is consistent with Fig. 4 : extreme allocations (i.e., very few plan updates or overly frequent updates) are generally suboptimal, whereas intermediate allocations better balance (i) asking a few related questions before updating the plan and (ii) replanning multiple times to correct early uncertainty.

**Takeaway.** Overall, replacing the question-agent backbone with GPT-5.2 preserves the main conclusion of the paper: *A-MPQC* consistently improves the alignment-burden tradeoff under matched budgets, with particularly strong gains in efficiency on banner generation, supporting that the benefits stem from the test-time control and format adaptation mechanisms rather than backbone-specific tuning.

*Table 5.* MIMO comparison with refusal-enabled baselines under the same respond-or-reject channel and cost accounting. The left block reports the original DG baselines, and the right block reports their refusal-enabled counterparts. "–" indicates that the measured similarity improvement is negative, so the efficiency ratio is not meaningful.

| Original Baselines | | | | Refusal-Enabled Baselines | | | |
|---|---|---|---|---|---|---|---|
| Method | $\tilde{C} \downarrow$ | $\Delta S \uparrow$ | $\Delta S/\tilde{C} \uparrow$ | Method | $\tilde{C} \downarrow$ | $\Delta S \uparrow$ | $\Delta S/\tilde{C} \uparrow$ |
| DG | 6.646 | 0.256 | 0.038 | DG + refusal | 1.857 | 0.308 | 0.166 |
| DG + Binary | 0.055 | 0.019 | 0.342 | DG + Binary + refusal | 0.321 | 0.008 | 0.025 |
| DG + Multiple-Choice | 1.614 | 0.070 | 0.043 | DG + Multiple-Choice + refusal | 0.311 | -0.017 | – |
| DG + Open-Ended | 5.240 | 0.209 | 0.040 | DG + Open-Ended + refusal | 2.801 | 0.035 | 0.012 |
| DG + Flexible | 2.318 | 0.457 | 0.197 | DG + Flexible + refusal | 1.410 | 0.411 | 0.291 |
| *A-MPQC* (Ours; respond-or-reject + rejection-based replanning) | | | | | 1.238 | 0.594 | **0.480** |

## A.3. Refusal-Enabled Baselines

The respond-or-reject interface in *A-MPQC* may appear to give the method an advantage because users can decline questions instead of always providing full answers. To examine this possibility, we additionally evaluate refusal-enabled versions of the DG baselines on MIMO. These baselines are given the same respond-or-reject channel and are evaluated with the same cost accounting as *A-MPQC*. This comparison separates two effects: whether refusal itself lowers interaction cost, and whether the agent can effectively use refusal feedback to improve the questioning policy.

Table 5 shows that enabling refusal indeed reduces the interaction cost of several baselines. For example, DG+Flexible decreases from $\tilde{C}=2.318$ to $\tilde{C}=1.410$, and its efficiency improves from 0.197 to 0.291. However, the improvement from refusal alone does not close the gap to *A-MPQC*, which still achieves the highest alignment improvement ($\Delta S=0.594$) and the best efficiency ($\Delta S/\tilde{C}=0.480$). Together with the *A-MPQC*-IgnoreReject ablation in Sec. 5.4, these results indicate that the main gain is not merely from allowing users to reject questions. Rather, *A-MPQC* benefits from using rejection as a surrogate feedback signal to replan subsequent questioning directions and formats.

# B. Implementation Details

This appendix provides implementation details, including (i) reusable prompt templates with verified input/output contracts, (ii) shared questioning principles, and (iii) evaluation prompts used by the LLM judge.

## B.1. Prompt Templates

The question agent policy (DG / *A-MPQC*), the user agent, the plan updater $U$, and the generator $G$. We enforce a strict **single-question protocol**: each interaction step asks exactly one question and receives exactly one reply (or rejection).

### B.1.1. SHARED PROTOCOL AND VARIABLES

**Interaction protocol.** All methods start from a shared initial design plan $d_0$ generated from `<P0>`. At each interaction step, the question agent produces exactly one query (DG) or one question plan (A-MPQC). The user agent returns exactly one response. For A-MPQC, the user agent answers only the `QUESTION` portion and ignores `NEXT`. After the satisfy check returns `SATISFIED`, we update the plan once using the plan updater $U$ and then perform single-shot generation from the final plan $d_n$ (no iterative refinement from intermediate outputs).

**Task-specific inputs.**

- **Banner case:** the user agent observes `<TARGET_IMAGE>` and `<LOGO_IMAGE>`.

- **Webpage case:** the user agent observes `<TARGET_WEBPAGE_IMAGE>` and `<TARGET_WEBPAGE_SUMMARY>`. The evaluation additionally provides `<ORIGINAL_ASSETS>` to the judge.

- **Assets:** `<ASSETS>` and `<ASSET_PATH_RULES>` are used when initializing the webpage plan and when judging asset usage.

**Variables.** We use the following placeholders in all prompt templates:

*Table 6.* Prompt usage sheet (roles, I/O contracts, and constraints).

| Block | Inputs / Role | Output / Constraints |
|---|---|---|
| **(0) Shared variables** | `<P0>` + `<PLAN>` + `<HISTORY>`.
Task-specific:
`<ASSETS>`/`<ASSET PATH RULES>` (webpage),
`<LOGO_IMAGE>` (banner). | N/A. |
| **(1) Question agent** (DG and DG variants) | **Role:** ask one clarification question to reduce uncertainty.
Inputs: `<P0>` + `<PLAN>` + `<HISTORY>`
(+ `<RETRIEVED TRACES>` for RAG). | **Output only ONE question.**
Format constraint optional: Binary / MCQ(A–C or A–D) / Open-ended / Flexible. |
| **(2) Question agent** (Proposed *A-MPQC*) | **Role:** same as DG, but additionally outputs a question plan with a follow-up direction.
Inputs: `<P0>` + `<PLAN>` + `<HISTORY>`. | **Output exactly ONE question plan line:**
`QUESTION: <ONE question>`
`| NEXT: <follow-up`
`direction>`.
**Only QUESTION is answered by the user; NEXT is not asked.** |
| **(3) User agent** | **Role:** simulate a user who can see the *true target artifact* and *hidden spec*; answer naturally.
Input: a question (DG) or a question plan (A-MPQC) plus an answering instruction. | Baselines: **answer-only** (output the answer text only).
*A-MPQC*: `respond:<answer>` **or** `reject: No`.
(When given a question plan, respond only to QUESTION and ignore NEXT.) |
| **(4) Plan updater** $U$ | **Role:** update the explicit plan using provided QA pairs.
Inputs: current `<PLAN>` + `<ACCEPTED_QA_PAIRS>`. | **Output only updated plan text.**
No invention; incorporate only provided QA pairs. |
| **(5) Generator** $G$ | **Role:** single-shot generation from the final plan.
Inputs: webpage: `<FINAL PLAN>`; banner: `<FINAL PLAN>` + `<LOGO_IMAGE>`. | Webpage: **complete HTML**.
Banner: **final image**. |
| **(6) Judge** | **Role:** evaluate similarity using a fixed rubric and return structured scores.
Inputs: target and generated renders (and original assets for webpage judging). | **Output:** five integer scores (1–5)
(+ structured feedback; we compute `average_score` for aggregation). |

(i) question agent emits one question (or a QUESTION | NEXT plan), (ii) user answers once, (iii) the plan is updated using provided QA pairs, (iv) generation is performed once from the final plan, and (v) evaluation returns five integer dimension scores.

- `<P0>`: initial user prompt $p_0$.
- `<PLAN>`: current plan summary $d_t$ (natural language).
- `<HISTORY>`: dialogue history summary.
- `<TARGET_IMAGE>`: target banner image $x^\star$ (user agent only).
- `<LOGO_IMAGE>`: logo image (banner case; user agent and generator).
- `<TARGET_WEBPAGE_IMAGE>`: target webpage screenshot/render (user agent only).
- `<TARGET_WEBPAGE_SUMMARY>`: short textual summary of the target webpage (user agent only).
- `<TARGET_SPEC>`: hidden requirement specification (user agent only).
- `<ASSETS>`: available assets (webpage case; plan initialization and user agent context).
- `<ASSET_PATH_RULES>`: allowed asset paths (webpage case).
- `<ORIGINAL_ASSETS>`: original assets provided to the webpage judge for asset verification.
- `<FINAL_PLAN>`: final plan $d_n$ used for generation.
- `<RETRIEVED_TRACES>`: retrieved exemplars (RAG baseline only).
- `<ACCEPTED_QA_PAIRS>`: QA pairs provided to the plan updater $U$.

### B.1.2. SHARED QUESTIONING PRINCIPLES

We reuse the question-asking principles articulated in (Hahn et al., 2025) to make the questioning behavior explicit and reproducible: *Relevance*, *Uncertainty Reduction*, *Easy-to-Answer*, and *No Redundancy*.

---

**Shared questioning principles (used for `<PRINCIPLES>`)**

```
 (1) Relevance:
- Base the question on the user prompt <P0> and the current plan <PLAN>.
- Ask only about constraints that can change the final artifact.
 (2) Uncertainty Reduction:
- Target the highest-uncertainty, highest-impact ambiguity first.
- Prefer questions whose answers will change or sharpen the plan.
 (3) Easy-to-Answer:
- Keep the question concise and direct.
- Provide enough context to reduce user understanding effort.
 (4) No Redundancy:
- Do NOT ask about information already present in <PLAN> or <HISTORY>.
- Do NOT re-ask rejected topics unless you narrow and rephrase.
 Hard constraints:
- Ask EXACTLY ONE question.
- Do NOT include explanations, analysis, or multiple questions in one turn.
```

---

### B.1.3. BASELINE QUESTION FORMAT

**Overview.** To standardize comparisons across baselines, we instantiate each baseline with an explicit `<QUESTION_FORMAT_INSTRUCTION>` that constrains the *form* of the next clarification question while keeping the rest of the prompting and backbone model unchanged. We consider four atomic formats: Direct Generation (DG), Binary, Multiple-Choice, and Open-Ended, and two composite variants that allow format selection (DG+Flexible) and additionally provide retrieved exemplars as guidance (DG+Flexible+RAG). All formats enforce a shared constraint of asking *exactly one* clarification question per round.

---

**Format: Directly Question Generation (used for `<QUESTION_FORMAT_INSTRUCTION>` )**

```
 Goal:  Ask ONE clarification question that maximally reduces uncertainty about the
 target artifact.
```

---

**Format: Binary Question Format (used for `<QUESTION_FORMAT_INSTRUCTION>` )**

```
 Goal:  Ask ONE binary question that maximally reduces uncertainty about the target
 artifact.
 Requirements:
- Ask a YES/NO question with a single proposition.
- Do NOT bundle multiple checks.
- Keep it answerable from the target image/spec.
 Examples:
- Should the CTA button be present?  (Yes/No)
- Should the hero section include a prominent CTA button?  (Yes/No)
```

---

**Format: Multiple-Choice Question Format (used for <QUESTION_FORMAT_INSTRUCTION> )**

```
Goal:  Ask ONE multiple-choice question that maximally reduces uncertainty about the
target artifact.
Requirements:
- Provide EXACTLY 3 or 4 options labeled (A)-(C) or (A)-(D).
- Options must be mutually exclusive and phrased consistently.
- The user answer must be one of:  (A), (B), (C) or (A), (B), (C), (D).
- Use the LAST option as "Other / none of the above" when appropriate to keep the
question easy-to-answer.
Output requirements:
- Output ONLY the question and options.
Examples:
(4-option)
- Which layout best matches the target?
(A) centered headline above CTA
(B) left-aligned text with right-side product image
(C) full-bleed image with bottom text overlay
(D) other / none of the above
(3-option)
- What is the primary hero visual style?
(A) large product photo
(B) illustration/mascot graphic
(C) other / none of the above
```

---

**Format: Open-Ended Question Format (used for <QUESTION_FORMAT_INSTRUCTION> )**

```
Goal:  Ask ONE open-ended question that maximally reduces uncertainty about the
target artifact.
Requirements:
- Ask for ONE short free-text answer.
- Make the requested information explicit (e.g., "exact headline text", "describe the
image in section 2").
- Do NOT ask the user to redesign; only to reveal the target intent.
Examples:
- What is the exact headline text shown on the banner?
- Briefly describe what image should appear in the second section (e.g., "team
photo", "dashboard screenshot").
```

---

**Format: DG+Flexible Question Format (used for <QUESTION_FORMAT_INSTRUCTION> )**

```
Goal:  Ask ONE clarification question that maximally reduces uncertainty about the
target artifact.
You may choose ONE of the following formats:
- Binary Question Format (YES/NO)
- Multiple-Choice Question Format with EXACTLY 3 or 4 options (A)-(C) or (A)-(D)
- Open-Ended Question Format (short free text)
Hard constraints:
- Ask EXACTLY ONE question.
- Follow the Shared questioning principles.
Examples:
- Example 1 (Binary):
Should the CTA button be present?  (Yes/No)
- Example 2 (Multiple-choice, 4 options):
Which layout best matches the target?
(A) centered headline above CTA
(B) left-aligned text with right-side product image
(C) full-bleed image with bottom text overlay
(D) other / none of the above
- Example 3 (Open-ended):
What is the exact headline text shown on the banner?
```

Format: DG+Flexible+RAG Question Format (used for <QUESTION_FORMAT_INSTRUCTION> )

```
Goal:  Ask ONE clarification question that maximally reduces uncertainty about the
target artifact.
You may choose ONE of the following formats:
- Binary Question Format (YES/NO)
- Multiple-Choice Question Format with EXACTLY 3 or 4 options (A)-(C) or (A)-(D)
- Open-Ended Question Format (short free text)
You will also be given retrieved questions from prior attempts on different
instances.
Use them ONLY as guidance for what to ask and which format to use.
Do NOT copy exemplar-specific content unless it clearly matches the current intent.
Retrieved exemplars (for reference):
<RETRIEVED_TRACES>
Hard constraints:
- Ask EXACTLY ONE question.
- Follow the Shared questioning principles.
Examples:
- Example 1 (Binary):
Should the CTA button be present?  (Yes/No)
- Example 2 (Multiple-choice, 4 options):
Which layout best matches the target?
(A) centered headline above CTA
(B) left-aligned text with right-side product image
(C) full-bleed image with bottom text overlay
(D) other / none of the above
- Example 3 (Open-ended):
What is the exact headline text shown on the banner?
```

### B.1.4. BASELINE QUESTION AGENT PROMPT TEMPLATE

Question agent system prompt (WebGen-V)

```
You are the question agent for webpage HTML generation.
Ask ONE clarification question to reduce uncertainty about the target webpage
requirements.
Follow the Shared questioning principles exactly.
<PRINCIPLE>
Focus on uncertain aspects such as:
layout/structure, visual style (colors/theme), typography/hierarchy, key
sections/content intent, and asset usage.
Output ONLY the question (no extra explanation).
```

Question agent system prompt (MIMO)

```
You are the question agent for banner image generation.
Ask ONE clarification question to reduce uncertainty about the target banner
requirements.
Follow the Shared questioning principles exactly.
<PRINCIPLE>
Focus on uncertain aspects such as:
message/CTA, brand tone and palette, layout hierarchy, typography feel, and logo
placement.
Output ONLY the question (no extra explanation).
```

Question agent user message

```
Initial prompt:  <P0>
Current plan:  <PLAN>
History:  <HISTORY>
<QUESTION_FORMAT_INSTRUCTION>
```

### B.1.5. A-MPQC QUESTION PLAN FORMAT

---

**Format: *A-MPQC* Question Plan**

```
Goal:  Propose ONE question plan that maximally reduces uncertainty about the target
artifact.
Output format:
QUESTION: <ONE question> | NEXT: <the follow-up direction in one sentence>
Notes:
- NEXT is NOT asked now.  It is a short description of what you intend to clarify
next if needed.
- NEXT should describe direction(s), not full questions.
- Choose ONE question format:  Binary / Multiple-Choice / Open-Ended.
You may choose ONE of the following formats:
- Binary Question Format (YES/NO)
- Multiple-Choice Question Format with EXACTLY 3 or 4 options (A)-(C) or (A)-(D)
- Open-Ended Question Format (short free text)
Examples:
- Example 1 (Binary):
QUESTION: Should the CTA button be present?  (Yes/No) | NEXT: confirm CTA style and
placement
- Example 2 (Multiple-choice, 4 options):
QUESTION: Which layout best matches the target?  (A) centered headline above CTA (B)
left-aligned text with right-side product image (C) full-bleed image with bottom
text overlay (D) other / none of the above | NEXT: confirm dominant palette and
refine typography
- Example 3 (Open-ended):
QUESTION: What is the exact headline text shown on the banner?  | NEXT: lock copy
details then verify a single layout constraint
```

---

**Question agent user message (A-MPQC; initial)**

```
Initial prompt:  <P0>
Current plan:  <PLAN>
History:  <HISTORY>
Instruction:  Output ONE question plan in the required QUESTION | NEXT format.
```

---

**Question agent user message (A-MPQC; replanning)**

```
Initial prompt:  <P0>
Updated plan (after last update):  <PLAN>
History:  <HISTORY>
Accepted question plans (answered; reinforce these directions):
<ACCEPTED_QUESTION_PLANS>
Rejected question plans (not answered; diagnose why and revise accordingly):
<REJECTED_QUESTION_PLANS>
Instruction:
Propose ONE new question plan in the required format:
QUESTION: <ONE question> | NEXT: <follow-up direction>
Follow the adaptive-format policy:
- Default to the least expressive format that you judge sufficient for the remaining
ambiguity:
Binary (yes/no) < Multiple-Choice (3--4 options) < Open-Ended (short free text).
- Increase expressiveness when a less expressive format was rejected because it
was too restrictive or did not allow the user to express the true target intent
(insufficient coverage).
Use accept/reject outcomes to update direction:
- For accepted plans:  do not re-ask the same information; treat the answered content
as resolved and move to the next highest-impact ambiguity.
- For rejected plans:  switch format according to the rule above.
```

B.1.6. USER AGENT PROMPT TEMPLATE

To reduce inference difficulty for webpage cases, we provide a brief textual summary along with the target webpage image, which may contain many small details. The user agent must answer using only the image, summary, and question. For banner cases, the agent receives only the target banner image and logo image, and answers using only these inputs plus the question.

---

Answer-Only Format (for baselines; used for `<ANSWER_FORMAT_INSTRUCTION>`)

```
Reply with ONLY the answer to the question.  Do not add any extra text, preface, or
explanation.
Rules:
- You will receive ONE question.
- Answer ONLY what is asked, concisely and naturally.
- Do NOT propose new questions.
- Do NOT volunteer extra details unless explicitly requested.
Examples:
Binary:  Yes
Multiple-choice:  (C)
Open-ended:  The headline is "Summer Sale".
```

---

Respond-or-Reject Format (for *A-MPQC*; used for `<ANSWER_FORMAT_INSTRUCTION>`)

```
You will receive ONE question plan in the form:
QUESTION: <one question> | NEXT: <follow-up direction>
Your job is to respond ONLY to the QUESTION part.
Ignore NEXT when answering; it is not being asked to you.
Reply with EXACTLY ONE of the following two lines:
- respond:  <your answer to QUESTION>
- reject:  No
Rules:
- This is a single-question interaction.  Answer ONLY what is asked in QUESTION.
- Keep the answer concise and natural.
- If the QUESTION cannot be answered from the provided inputs, output exactly:
reject:  No
- Do NOT add any other text.
Examples:
Binary:
QUESTION: Should the CTA button be present?  (Yes/No) | NEXT: confirm CTA style
respond:  Yes
Multiple-choice:
QUESTION: Which layout best matches the target?  (A) centered headline above CTA (B)
left-aligned text with right-side product image (C) full-bleed image with bottom
text overlay (D) other / none of the above | NEXT: refine typography
respond:  (C)
Open-ended:
QUESTION: What is the exact headline text shown on the banner?  | NEXT: lock copy
details
respond:  The headline is "Summer Sale".
Reject:
QUESTION: What is the brand's internal tagline?  | NEXT: refine tone
reject:  No
```

---

User agent system prompt (shared)

```
You are a human collaborator answering clarification questions.
You will be shown the ground-truth target inputs and ONE question.
Your job:
- Answer ONLY the question you receive.
- Do NOT propose new questions.
- Do NOT volunteer extra details unless explicitly requested.
You must follow the answer format specified in the instruction for this run.
```

---

**User agent user message (webpage)**

```
Target webpage image:  <TARGET_WEBPAGE_IMAGE>
Target webpage summary:  <TARGET_WEBPAGE_SUMMARY>
Answer format instruction:  <ANSWER_FORMAT_INSTRUCTION>
Question:  <QUESTION from Agents>
```

---

**User agent user message (banner)**

```
Target image:  <TARGET_IMAGE>
Logo image:  <LOGO_IMAGE>
Answer format instruction:  <ANSWER_FORMAT_INSTRUCTION>
Question:  <QUESTION from Agents>
```

---

### B.1.7. INITIAL PLAN GENERATOR, PLAN UPDATER, AND GENERATORS

**Shared initial design plan.** All methods share the same initial design plan $d_0$, which is generated once from the initial user prompt `<P0>` (and task-provided assets when applicable) before any interaction begins.

---

**Initial design plan generator (webpage case; JSON plan)**

```
You are a professional web designer creating an initial design plan based on a
client's request.
CLIENT'S REQUEST: <P0>
AVAILABLE ASSETS (these are the exact filenames you can use):  <ASSETS>
YOUR TASK:
Create a COMPLETE initial design plan in JSON format.  Make reasonable design
decisions based on the client's request - don't leave anything blank.  The Q&A
process will refine and correct your assumptions.
REQUIRED JSON STRUCTURE:

{
  "global_style": {
    "overall_mood": "professional/modern/friendly/etc based on request",
    "color_palette": "describe primary and accent colors",
    "typography_feel": "clean sans-serif/elegant serif/etc"
  },
  "available_assets": ["list", "of", "asset", "filenames.png"],
  "sections": [
    {
      "order": 1,
      "type": "hero",
      "goal": "Brief description of section purpose",
      "layout": "centered/left-aligned/split layout description",
      "colors": "background and text colors",
      "columns": "1/2/3/4 column layout",
      "alignment": "left/center/right",
      "image_asset": "specific filename from available_assets",
      "image_position": "left/right/center/background",
      "elements": ["heading", "subtext", "cta_button"]
    }
  ]
}

GUIDELINES:
1.  Infer sections from the client's request (hero, features, testimonials, etc.)
2.  Copy ALL available assets into "available_assets" with exact filenames
3.  Make SPECIFIC design decisions - no placeholders
4.  Assign images from "available_assets" to appropriate sections
5.  Specify layout details:  columns, alignment, image positions
6.  The plan should be detailed enough to generate HTML directly
Output valid JSON only.
```

---

Initial design plan generator (banner case; JSON plan)

```
You are a professional visual designer creating an initial banner design plan based
on a client's request.
CLIENT'S REQUEST:
<P0>
LOGO IMAGE:
<LOGO_IMAGE>
YOUR TASK:
Create a COMPLETE initial banner design plan in JSON format.  Make reasonable design
decisions based on the client's request - don't leave anything blank.  The Q&A
process will refine and correct your assumptions.
REQUIRED JSON STRUCTURE (example schema):

{
  "design_plan": {
    "background_image_plan": {
      "description": "Describe the background imagery in detail.",
      "style": "Overall visual style.",
      "colors": "Background palette and accent colors.",
      "mood": "Emotional tone.",
      "composition": "High-level composition and negative space."
    },
    "ad_copy": {
      "headline": "Main headline text",
      "description": "Short supporting description",
      "cta_text": "CTA button text"
    },
    "color_scheme": {
      "primary_colors": "Primary colors and accents",
      "tone": "Overall tone (cool/warm/etc.)",
      "contrast_requirements": "Text/CTA contrast requirements"
    },
    "style_and_mood": {
      "overall_style": "Modern/Minimalist/etc.",
      "emotional_tone": "Brand emotion to convey"
    },
    "layout": {
      "text_placement": "Where the text block goes and alignment",
      "logo_placement": "Where the logo goes",
      "cta_placement": "Where the CTA goes",
      "logo_size": "Approximate size guidance",
      "headline_size": "Relative size guidance",
      "description_size": "Relative size guidance",
      "cta_size": "Relative size guidance",
      "composition_approach": "Overall layout approach (e.g., asymmetrical, two-column)"
    }
  }
}
GUIDELINES:
1.  Infer a clear layout from the client's request (hierarchy:  logo, headline,
description, CTA)
2.  Make SPECIFIC decisions (palette, typography feel, composition, spacing)
3.  Ensure high readability (contrast, text placement, CTA visibility)
4.  Ensure the plan is detailed enough for image generation
Output valid JSON only.
```

**Plan updater.** All methods use the same frozen plan updater $U$. After each interaction step, $U$ updates the current plan $d_t$ into $d_{t+1}$ using the available QA pairs. For our method, only *accepted* QA pairs are provided to $U$ (rejected items are excluded). For baselines without rejection, all QA pairs are treated as accepted. This design ensures that differences across methods arise from questioning policies rather than from different update/generation components.

Plan updater system prompt (U; shared)

```
You are the design plan updater.
Given the current plan and the accepted QA pairs, update the plan to reflect the
newly specified constraints.
Rules:
- Preserve constraints already specified unless explicitly contradicted by an
accepted answer.
- Add new constraints ONLY from the accepted QA pairs provided.
- Keep the plan concise and structured.
- Do not invent details not supported by the provided QA pairs.
Output ONLY the updated plan text (no extra explanation).
```

Plan updater user message (U; shared)

```
Current plan:  <PLAN>
Accepted QA pairs:  <ACCEPTED_QA_PAIRS>
Instruction:  Output the updated plan in same structure.
```

**Generators (shared across methods).** After interaction terminates, we perform single-shot generation from the final plan $d_n$. All methods use the same generators $G$ for each task, and generation is executed once. For webpage generation we follow the original benchmark protocol and use the same generator implementation described in the benchmark documentation. For banner generation we likewise follow the benchmark protocol and use the same image generator implementation.

### B.1.8. SATISFY CHECK (SHARED STOP CRITERION)

**Purpose.** Before asking a new question, all methods run a satisfy check to decide whether further clarification is necessary. If the check returns SATISFIED, the interaction terminates and the generator is invoked once using the final plan.

Satisfy check system prompt (shared)

```
You are the question agent for visual design intent elicitation.
Follow the Shared questioning principles exactly.
Task:
Given the current plan and dialogue history, decide whether the plan is sufficiently
specified to proceed to final generation WITHOUT asking more questions.
Output EXACTLY one of:
- SATISFIED
- CONTINUE
Rules:
- If all major constraints needed for generation are already specified in current
plan (and no unresolved ambiguities remain in QA history), output SATISFIED.
- If any high-impact ambiguity remains that is likely to change the artifact (e.g.,
missing key copy, missing layout choice, unclear imagery/asset usage, unresolved
color/style decisions), output CONTINUE.
- Do NOT propose a question here.
- Do NOT add any other text.
```

Satisfy check user message (shared)

```
Current plan:  <PLAN>
History:  <HISTORY>
Instruction:  Output SATISFIED or CONTINUE.
```

### B.2. Retrieval Baseline Details

We build an offline bank of interaction traces. Each trace contains: (i) design plan, (ii) QA pairs, and (iii) the resulting similarity improvement. We embed each serialized trace using all-MiniLM-L6-v2. At test time, we form the retrieval query from the current context only (<PLAN> + <HISTORY>) and retrieve the top-$k$ most relevant traces (we use $k=3$). The retrieved exemplars are appended verbatim as <RETRIEVED_TRACES> to the DG+Flexible+RAG prompt as reference.

## B.3. Evaluation Prompts

---

**Judge system prompt (Webpage similarity; JSON output)**

You are an expert evaluator assessing how well a **generated webpage** matches a **reference webpage**.  This is a rigorous evaluation for academic research purposes.
**INPUTS PROVIDED** 1.  **Generated Webpage Screenshot** --- The webpage produced by the generation system 2.  **Reference Webpage Screenshot** --- The ground truth target webpage 3.  **Original Assets** --- The source images/icons that should be used in the generated webpage
**EVALUATION APPROACH**
**Key Principle:  Section-by-Section Analysis**
Webpages are composed of multiple **sections** (e.g., hero, features, testimonials, pricing, footer).  Evaluate each section's fidelity rather than treating the page as a single monolithic image.
**Asset Usage Verification**
Compare the provided assets against what appears in the generated webpage.  Check whether correct assets are used, placed in correct positions, sized properly, and not incorrectly substituted.
**FIVE EVALUATION DIMENSIONS**
1.  **Layout_Structure (Section Arrangement & Grid)**
2.  **Image_Usage (Asset Placement & Selection)**
3.  **Color_Scheme (Palette & Mood)**
4.  **Typography (Font Choices & Hierarchy)**
5.  **Section_Completeness (Content Fidelity per Section)**
Use the rubric strictly; a score of 5 means near-identical.
**OUTPUT FORMAT**
Provide your evaluation as valid JSON:

```
{
    "section_analysis": {
        "identified_sections": ["hero", "features", "testimonials", "footer"],
        "section_match_rate": "X/Y sections correctly structured"
    },
    "asset_analysis": {
        "assets_in_reference": ["list of visible assets"],
        "assets_correctly_used": ["list of correctly placed assets"],
        "assets_missing_or_wrong": ["list of issues"]
    },
    "dimension_scores": {
        "Layout_Structure": {
            "score": <1-5>,
            "reasoning": "<specific observations about section arrangement>"
        },
        "Image_Usage": {
            "score": <1-5>,
            "reasoning": "<specific observations about asset placement>"
        },
        "Color_Scheme": {
            "score": <1-5>,
            "reasoning": "<specific observations about colors>"
        },
        "Typography": {
            "score": <1-5>,
            "reasoning": "<specific observations about fonts>"
        },
        "Section_Completeness": {
            "score": <1-5>,
            "reasoning": "<specific observations about content fidelity>"
        }
    }
}
```

Now evaluate the provided screenshots and assets:

---

---

**Judge system prompt (Banner similarity; JSON output)**

You are an expert visual design evaluator. Evaluate similarity between a generated
banner and a reference banner across five dimensions, using integer scores 1--5.
Two images provided:
1. **Reference Image:** Ground truth banner design
2. **Generated Image:** Generated banner image to evaluate
Assess visual similarity, not quality. Match the reference exactly.
**Scoring Scale**
**1 (Completely Different):** No meaningful similarity. Fundamental differences in
structure, elements, or appearance.
**2 (Mostly Different):** Minimal similarity. Only 1--2 minor shared characteristics.
Majority of elements differ.
**3 (Moderately Similar):** Partial similarity. ~30--50% visual characteristics match.
Some elements align, but key differences remain.
**4 (Very Similar):** High similarity. ~70--80% visual characteristics match. Most
elements closely align with minor variations.
**5 (Nearly Identical):** Near-perfect similarity. 90%+ visual characteristics match.
Elements nearly identical in appearance, placement, and content.
**Dimensions**
1. **Overall_Color:** Color scheme similarity – dominant colors, palette composition,
gradients, color harmony. Primary color matches (background, dominant colors);
secondary/accent matches; palette composition; warm/cool relationships; gradient
similarity (direction/stops/colors); overall color mood and tone. 1 = fundamentally
different; 2 = mostly different; 3 = partial similarity (30--50% shared palette); 4
= high similarity (70--80% shared palette); 5 = nearly identical.
2. **Layout_Composition:** Element arrangement similarity – spatial organization,
positioning, alignment, structural composition. This dimension requires strict
evaluation. Layout structure similarity is fundamental. Verify all critical
elements present in reference: logo, headline, description/body, CTA button,
decorative/graphic elements. Missing critical element (logo, CTA, headline,
description) → maximum score 2. Elements in fundamentally different positions →
maximum score 2. Different layout structure → maximum score 3. Only color/text
similarity but completely different layout → score 1--2.
3. **Button_Style:** CTA button similarity – presence, position, shape, size, color,
text, styling (border/shadow/gradient/outline), corner radius, and effects. Missing
button entirely → score 1.
Fundamentally different position/shape/color → maximum score 2. Different styling
category (flat vs 3D, outlined vs filled, gradient vs solid) → maximum score 3.
4. **Typography_Hierarchy:** Typographic similarity – font characteristics, text
styling, size relationships, hierarchy, readability. Text occlusion/illegibility
or inverted hierarchy or poor contrast → maximum score 2.
5. **Image_Content:** Visual imagery similarity – background images/illustrations/graphics,
decorative elements, style category and composition.
6. **Text_Content:** Textual content similarity – exact wording of
headline/description/CTA and other text elements.
**Critical Failure Conditions:** Apply mandatory penalties for missing critical
elements, fundamental layout mismatch, missing button, fundamental button mismatch,
text occlusion, and illegible typography.
**Output Format**
Return JSON only (no markdown, no additional text):

```
{
  "scores": {
    "Overall_Color": <integer 1-5>,
    "Layout_Composition": <integer 1-5>,
    "Button_Style": <integer 1-5>,
    "Image_Content": <integer 1-5>,
    "Text_Content": <integer 1-5>
  },
  "feedback": {
    "Overall_Color": "<brief assessment, 1-2 sentences>",
    "Layout_Composition": "<brief assessment, 1-2 sentences>",
    "Button_Style": "<brief assessment, 1-2 sentences>",
    "Image_Content": "<brief assessment, 1-2 sentences>",
    "Text_Content": "<brief assessment, 1-2 sentences>"
  }
}
```

# C. Qualitative Examples

This section provides qualitative comparisons that complement the quantitative results in Sec. 5. In particular, WebGen-V emphasizes section-level alignment on long webpages (where local errors can hide under similar global structure), while MIMO (banner) makes element-level mismatches and hallucinations more visually salient. We also include an illustrative question-convergence example to clarify how *A-MPQC* adapts question formats over replanning rounds.

## C.1. WebGen-V Examples

Compared to banner generation, webpage generation contains substantially more heterogeneous elements (multiple sections, repeated layout motifs, and image/text alignment constraints) that often require clarification. In practice, the most consequential differences across methods frequently appear within sections: e.g., which image is used, whether it matches the intended content, whether the corresponding caption/CTA is aligned, and how spacing/hierarchy is expressed, rather than in the overall page skeleton.

Moreover, because all methods share the same initial design plan, the global page structure tends not to diverge drastically; the more informative comparison is therefore *section-local* changes that accumulate across the page.

This motivates using WebGen-V: it provides section-level renders/crops that make fine-grained discrepancies visible. At the same time, because evaluation aggregates scores across sections, localized gains (or failures) can be partially averaged out in a single overall similarity number.

Figure 7 shows a representative instance with the target webpage (left), DG (middle), and *A-MPQC* (right). In this setting, *A-MPQC* tends to (i) lock in high-level layout decisions early and (ii) spend subsequent budget on section-level ambiguities that matter most for perceived alignment (e.g., correcting image usage and tightening text/image alignment within the same section), rather than repeatedly revisiting already-settled global choices. Because section-level deviations can propagate (e.g., a mistaken hero assumption affects downstream styling and spacing), targeted replanning and format adaptation are particularly beneficial for long webpages.

## C.2. MIMO Examples

In banner generation, there are fewer distinct regions and fewer elements that *need* clarification. As a result, qualitative differences are often more immediate: whether the method captured the correct key elements (logo usage, headline/CTA content, layout template, and color palette), and whether it avoided hallucinating extra objects. This makes MIMO a useful complement to WebGen-V: it highlights element-level faithfulness, and failures without user interaction are often visually obvious (e.g., hallucinated imagery or incorrect composition), leading to a clear mismatch from the target.

Fig. 8 compares representative banners. With no user interaction, the generator can commit to plausible-but-incorrect assumptions (hallucinated elements or wrong layout archetype), yielding a noticeable gap to the ground truth. By contrast, *A-MPQC* more reliably captures the key categorical decisions (e.g., dominant color theme, layout template, logo position) and then refines remaining details, producing a closer match. Because banners have fewer sections to "average out" errors, qualitative improvements are often aligned with quantitative gains.

## C.3. Question Convergence Examples

Fig. 6 shows a representative *A-MPQC* run on a banner instance with a small budget scheduled as $B = n \cdot m$ using $n{=}3$ replanning rounds and up to $m{=}3$ questions per round (maximum $B{=}9$). The figure visualizes the full interaction loop: starting from a shared initial design plan, the agent asks clarification questions, updates the design plan using *accepted* answers only, and generates an intermediate banner from the updated plan to monitor convergence toward the target. The intermediate generations make the convergence effect tangible: after each plan update, the banner moves closer to the target in composition, imagery, and copy.

A key detail is that the questioning trajectory is not a fixed "open-ended $\rightarrow$ multiple-choice $\rightarrow$ binary" template. Instead, *A-MPQC* adapts to user behavior and its own confidence. In this example, the agent initially attempts low-burden multiple-choice questions to quickly disambiguate high-level intent, but early questions are rejected. After rejection, it backs off to a more answerable query, enabling a first plan update. In later rounds, once the overall design direction is established, the agent uses open-ended questions to elicit missing high-impact specifics (e.g., concrete imagery and exact copy), while still switching to binary or multiple-choice when a single verification or a small set of options is sufficient.

Finally, *A-MPQC* does not necessarily exhaust the allocated budget. After Round 2, the intermediate banner is already highly similar to the target, so the agent terminates early rather than spending additional questions on marginal refinements. This illustrates the budget-aware nature of the method: it allocates question effort where it yields the largest plan correction, and stops when additional clarification is unlikely to improve the outcome.

**Takeaway.** This example highlights that *A-MPQC* improves generation via an adaptive process: questions are chosen to maximize plan correction, formats are adapted (including backing off to open-ended after rejection and using binary/MCQ when confident), and early stopping prevents spending user effort on low-return refinements once the artifact has converged.

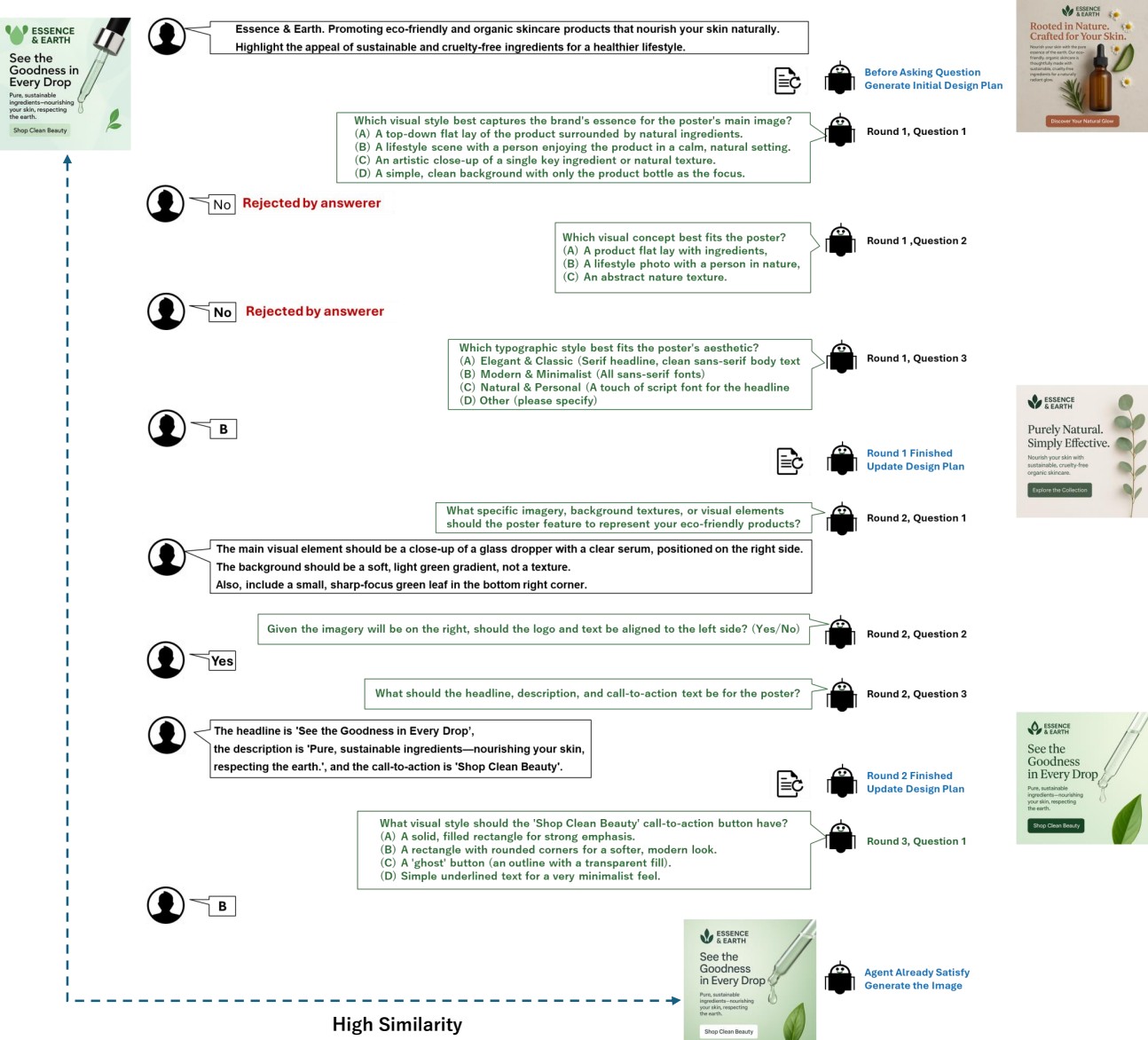

*Figure 6.* **Question convergence of *A-MPQC* on a banner instance** ($n=3$, $m=3$). The agent iteratively asks questions, updates the design plan from accepted answers, and generates intermediate banners that progressively approach the target. After sufficient similarity is reached, it early-stops without using the full budget.

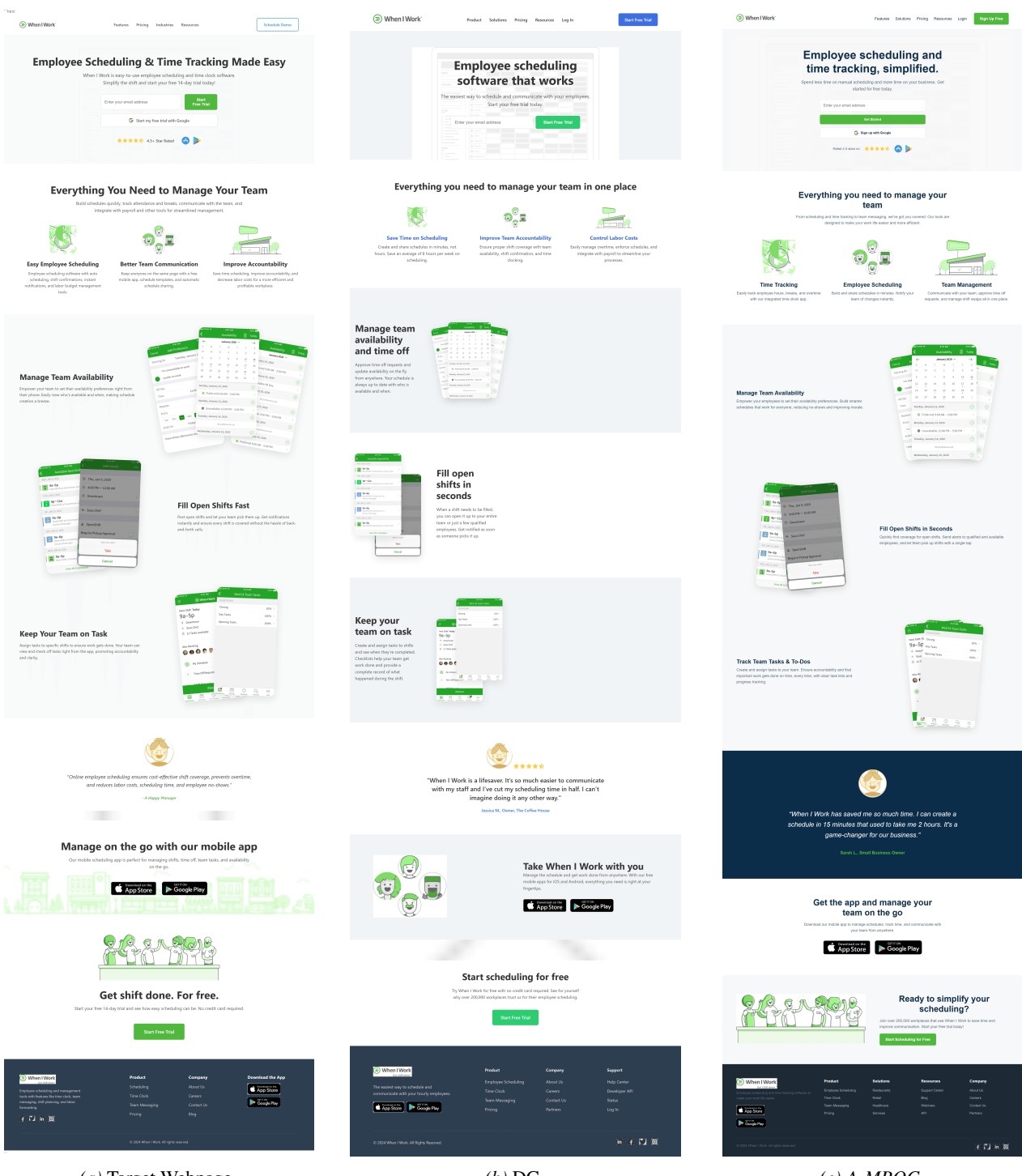

*(a)* Target Webpage.                *(b)* DG.                *(c)* A-MPQC.

*Figure 7.* WebGen-V qualitative comparison: target (left), DG (middle), and *A-MPQC* (right), shown at the same vertical scale.

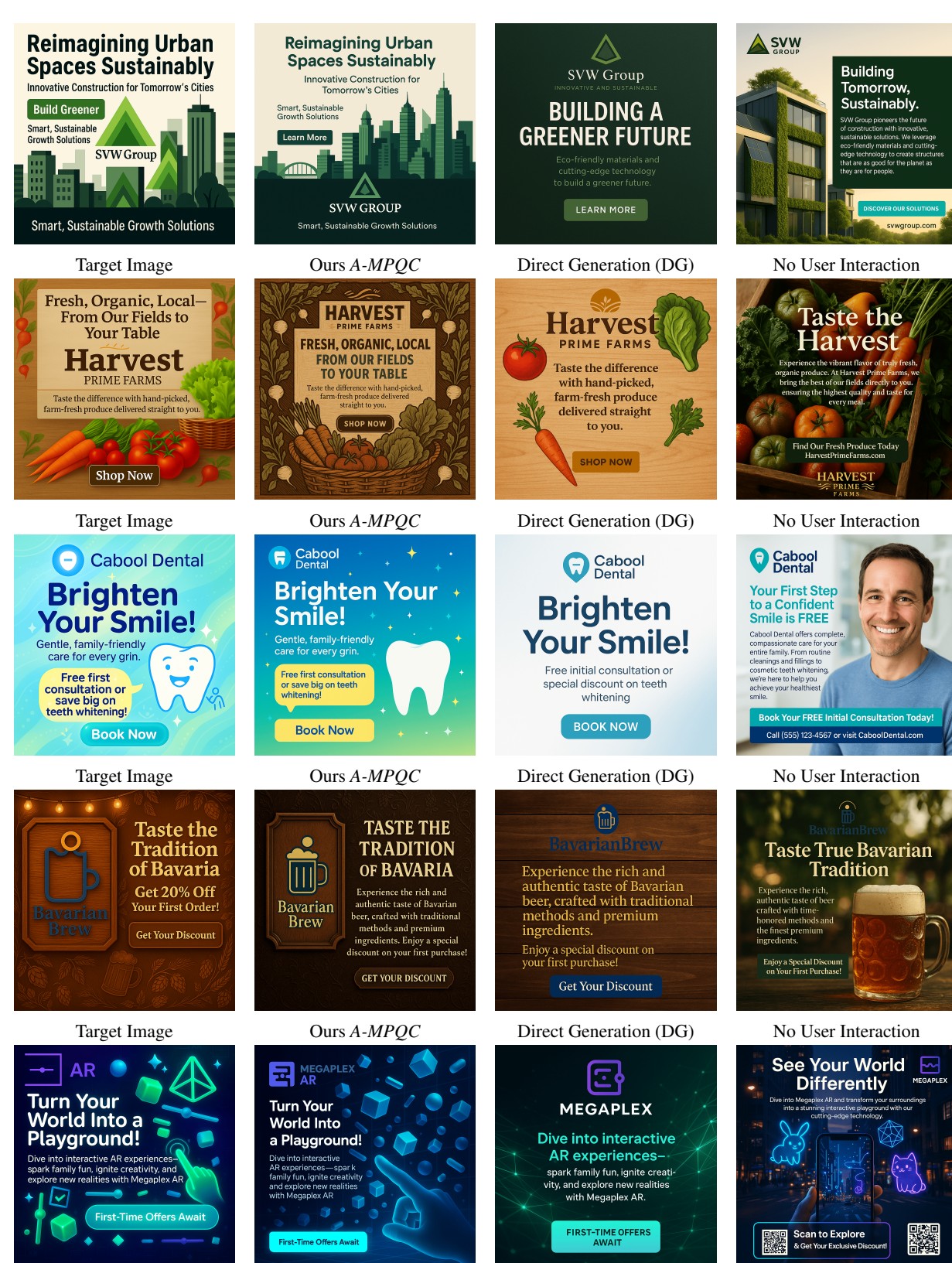

*Figure 8.* MIMO qualitative comparisons across banner styles: target image, *A-MPQC*, DG, and No User Interaction baseline.

