# OpenReview forum: "Agentic Model Predictive Questioning Control in Visual Design"
_ICML.cc/2026/Conference — ICML 2026 regular_

### Official Review · Reviewer_t7e5 · 2026-03-07

**Soundness:** 3
**Presentation:** 2
**Significance:** 3
**Originality:** 2
**Overall Recommendation:** 4
**Confidence:** 4

**Summary:**

This paper focuses on improving human-AI interaction in visual design tasks by balancing the tradeoff between user cognitive cost and generation quality. The proposed method, A-MPQC, is a training-free, test-time inference pipeline that optimizes the interaction trajectory by formulating multi-round clarification as a trajectory optimization problem. The framework is built upon three key designs: lookahead question plans to reduce ambiguity early, a respond-or-reject surrogate reward to steer querying, and adaptive question formats to further minimize user burden. Experiments conducted on webpage and ad banner generation benchmarks demonstrate that A-MPQC not only generates designs better aligned with user intent, but also achieves lower user-interaction costs across diverse interaction baselines.

**Compliance With Llm Reviewing Policy:**

Affirmed.

**Final Justification:**

Some of my concerns have been adequately addressed. I have updated my review accordingly.

**Key Questions For Authors:**

1. Have you tried A-MPQC on open-sourced LLMs? What would their performance be like?

2. How does the token-based proxy for human cognitive burden strictly correlate with actual human effort in this specific visual design context, and are there preliminary results from real human-user studies?

3. Given the reliance on LLM-as-a-judge for evaluating design similarity, how do you account for potential evaluation bias, and have you conducted human baseline comparisons for these judgments?

**Limitations:**

yes

**Strengths And Weaknesses:**

Strengths

1. The motivation is strong and highly practical. Besides the quality of LLM-assited visual design (or intent alignment), cognitive burden is also of great importance in terms of human-AI collaboration.

2. A-MPQC provides a effective and efficient perspective on improving proactive LLM agent system without retraining a large model.

3. Experiments on webpage and ad banner generation benchmarks show performance improvement. Different models are tested to demonstrate the robustness of the protocal. Comparisions and ablations are rigorous.

Weaknesses

1. The paper's technical contribution is somewhat limited, as it relies entirely on inference-time prompt engineering. Although the multi-round clarification process is heavily formalized, the paper to some extent lacks an algorithmic design or implementation with sufficient insight.

2. The formalization using MPC appears overly heavy and loosely connected to the actual implementation.

3. The methodology may be problematic. LLM token consumption are  used as a direct proxy for human cognitive without sufficient validation under the setting of human-AI interaction and visual design. LLM-based simulation is an imperfect approximation that should not substitute for actual human testing. Additional user studies may be helpful to improve the soundness of the paper's contributions.

---

> ### Author Rebuttal · Authors · 2026-03-30
>
> We thank the reviewer for the thoughtful and constructive feedback. We address each concern point-by-point below.
>
> > [W1] The paper's technical contribution is somewhat limited, as it relies entirely on inference-time prompt engineering.
>
> While the implementation is prompt-based, the contribution is not about prompt wording itself. The paper studies an underexplored problem: how question format affects user burden and final alignment in multi-round clarification, and how to optimize this tradeoff under a fixed interaction budget. We use an inference-time framework because design preferences are highly user- and task-dependent, making large-scale supervision for a learned policy expensive and hard to generalize (cf. Ln 86). Moreover, our experiments separate the effect of the proposed designs from ordinary prompting: DG + Flexible is already a strong prompting baseline, while A-MPQC adds three proposed designs and consistently improves over it. We therefore view the contribution as a burden-aware interaction framework, rather than prompt engineering alone.
>
> > [W2] The formalization using MPC appears overly heavy and loosely connected to the actual implementation.
>
> Our claim is not that we implement a full standard MPC. Rather, A-MPQC is an interaction-level MPC formulation: the design plan is the state, the questioning decision is the action, respond/reject is a low-cost surrogate reward, and replanning occurs after each plan update. This is directly reflected in the implementation through our three designs.
>
> > [Q1] Have you tried A-MPQC on open-sourced LLMs? What would their performance be like?
>
> We do not include open-source MLLMs in the current paper because the user agent in Sec. 3.1 is assumed to be accurate with respect to the target design, and the open-source models we tested were not reliable enough for that role. The paper already includes cross-model results with GPT-5 as the question agent and Gemini-2.5-Pro as the user agent, and our additional experiments with GPT-5 as both the user agent and question agent lead to the same qualitative conclusion. Please see Reviewer Afw1 (Q2/Q4) and (W2/Q3) for the added results.
>
> > [W3,Q2] Additional user studies may be helpful to improve the soundness of the paper's contributions.
>
> We conducted a small-scale human validation study on the same multi-turn Q&A interactions used in our LLM-user simulations. Specifically, six LLM experts reviewed and rated a total of 687 interaction instances from both our method and the baselines. Each rater was assigned a disjoint subset of brands/conditions and evaluated each instance on a 1–5 scale for (i) reasoning demand and (ii) output demand, based on the displayed questions, answers, and token summaries. We then compared these human ratings with the logged reasoning/output token counts and our token-based burden proxy.
>
> Since our goal is to examine whether the token-based proxy preserves the relative ordering of perceived burden, rather than matching absolute scores, we report rank-based correlations: **Spearman’s ρ and Kendall’s τ**.
>
> | Human-rated dimension | Proxy / logged signal | Spearman’s ρ | Kendall’s τ |
> |---|---|---:|---:|
> | Output burden | Output tokens | 0.71 | 0.49 |
> | Reasoning burden | Reasoning tokens | 0.41 | 0.28 |
> | Combined burden (reasoning + output) | log(1 + token burden proxy) | 0.65 | 0.48 |
>
> These results suggest that the token-based proxy shows positive alignment with human-perceived interaction burden. Agreement on the reasoning side is weaker but remains directionally consistent. This is likely because perceived reasoning burden is inherently harder to discriminate consistently than output burden, particularly with a coarse ordinal 1–5 scale that produces many tied ratings.
>
> Overall, this small-scale human study provides preliminary evidence that our interaction-cost proxy captures meaningful aspects of perceived burden, while also suggesting that reasoning burden is inherently harder to measure reliably with coarse subjective ratings. We will include this validation in the final manuscript, and view it as an initial step toward a larger-scale study with more real-world users.
>
> > [Q3] Have you conducted human baseline comparisons for these judgments?
>
> We follow the original benchmark protocols for both WebGen-V and MIMO, as stated in Sec. 5.1.3. However, some bias may still remain. We conducted a small human comparison study in which human annotators rated output–target similarity on the same 1–5 scale used in our evaluation setting, and we then compared the resulting human ranking with the ranking induced by the benchmark judge on matched instances. We found strong rank agreement, with **Spearman’s $\rho$ = 0.81** and **Kendall’s $\tau$ = 0.59**, providing initial evidence that the benchmark similarity evaluation is reasonably aligned with human judgment in this setting. We will report these results conservatively in the final manuscript, as the sample size is still limited.

---

> > ### Author Rebuttal · Reviewer_t7e5 · 2026-04-02
> >
> > Thank you for your clarification. Here are some of the follow-up questions for the authors.
> >
> > 1. The primary human validation study relies on post-hoc evaluation rather than real-time interaction, which may fail to reflect the real user experiences of the interactive system.
> >
> > 2. The justification for omitting open-source models lacks empirical rigor. A failure analysis or quantitative evidences can better support this claim.
> >
> > 3. The formalization using MPC appears still an unnecessary over-formalization that misrepresents the actual contributions of the implementation
> >
> > Some of my concerns have been adequately addressed. I have updated my review accordingly.

---

> > > ### Author Response · Authors · 2026-04-03
> > >
> > > Thank you for your follow-up and for updating your review. We sincerely appreciate the time and effort you have invested in carefully reading our rebuttal and manuscript. We are glad that some of your concerns have been addressed, and we would like to clarify the remaining points below.
> > >
> > > ### 1. On the human validation study being “post-hoc” rather than real-time
> > > We respectfully believe there may be a misunderstanding here. Our human validation study is **not post-hoc** in the sense of asking annotators to retrospectively judge a completed interaction trajectory after the fact. Rather, the evaluation is conducted **step-by-step during the interaction process itself**.
> > >
> > > More concretely, at each round, the human participant is shown the **actual question generated by the Question Agent at that step** and is asked to rate how difficult that question would be for them to answer on a 1–5 scale. We then aggregate these per-step ratings across the full interaction trajectory and across participants. Therefore, the study is designed to measure the **real-time burden of answering each intermediate question**, rather than a retrospective judgment of the completed dialogue.
> > > We agree that this distinction was not sufficiently explicit in the current manuscript, and we will revise the text to clarify the protocol more clearly.
> > >
> > > ### 2. On the omission of open-source models
> > > We appreciate this concern and agree that the point should be supported empirically. To address this, we did additional experiment for banner generation task, and add one row in our result table under the same A-MPQC protocol, using **Qwen-VL** for both the **question**  module, while keeping **GPT** fixed as the **answer (user) agent**, so that user-side grounding remains strong. The evaluation metrics are the same as in the paper: **C** denotes total token cost, **ΔS** denotes similarity gain over DG, and **ΔS / C** denotes gain per unit cost.
> > >
> > > | Method | C ↓ | ΔS ↑ | ΔS / C ↑ |
> > > |--------|-----|------|----------|
> > > | DG | 6.55 | 0.41 | 0.06 |
> > > | DG + Binary | 0.05 | 0.00 | 0.00 |
> > > | DG + Multiple-Choice | 1.36 | 0.08 | 0.06 |
> > > | DG + Open-Ended | 6.79 | 0.18 | 0.03 |
> > > | DG + Flexible | 3.17 | 0.46 | 0.15 |
> > > | DG + Flexible + RAG | 2.39 | 0.35 | 0.15 |
> > > | A-MPQC (ours), GPT Q+Gen / GPT Ans. | 0.23 | 0.39 | 1.69 |
> > > | A-MPQC, Qwen-VL Q+Gen / GPT Ans. | 0.27 | 0.18 | 0.67 |
> > >
> > > These results clarify our design choice. While the cheaper backbone reduces cost, it also substantially lowers **ΔS**, i.e., the final output quality improvement. Therefore, lower cost **does not necessarily imply** a better method, and **ΔS / C** does not always improve when the generator is too weak.
> > >
> > > This is why we did not center the paper on open-source backbones in our original manuscript. The main goal of this work is to study **how an agent should interact with a human to optimize the trade-off between token cost and final accuracy**. When the generator is too weak, final performance becomes bottlenecked by the backbone itself, making it difficult to isolate the effect of the interaction strategy. In the revised manuscript, we will add representative failure cases and clarify that, although our interaction strategy can reduce unnecessary cost, strong MLLMs remain important when high-quality generation is required.
> > > ### 3. On the concern that the MPC formalization is an unnecessary over-formalization
> > > We respectfully disagree with this characterization.The attached implementation code is fully consistent with the method described in the paper, showing that the formulation is not an abstract over-formalization, but the actual principle guiding the algorithm design.
> > >
> > > More importantly, we have provided empirical support for the usefulness of this formulation. In both the **ablation studies in the main paper** and the additional analyses discussed in the rebuttal (particularly in response to Reviewer mAtH, W4), we included extensive experiments validating the effectiveness of the key components motivated by this formulation. These results show that the proposed design is not merely a formal abstraction, but is tied to measurable performance gains.
> > >
> > > That said, we acknowledge that the current presentation may have made the formalism appear heavier than necessary. In the revised manuscript, we will improve the exposition to more clearly distinguish:
> > > - the **core practical contribution**,
> > > - the **algorithmic intuition** behind sequential question planning, and
> > > - the **role of the MPC-inspired formulation** as a principled way to organize the method.
> > >
> > > Again, we sincerely thank you for the constructive feedback. We hope our final reply addresses all the concerns you raised.
> > > A warm and big THANK YOU!

---

### Official Review · Reviewer_kbxB · 2026-03-08

**Soundness:** 2
**Presentation:** 3
**Significance:** 3
**Originality:** 3
**Overall Recommendation:** 4
**Confidence:** 3

**Summary:**

The manuscript studies a pressing issue in interactive visual generation: how to ask clarification questions that improve alignment with user intent while also accounting for the user’s interaction burden. The research's principal concept is to cast multi-round clarification as a test-time trajectory optimization problem under a fixed question budget, using a model-predictive-control-style framework called A-MPQC. The method maintains an explicit design plan, proposes question plans with short-horizon lookahead, uses a respond-or-reject mechanism as a surrogate reward for replanning, and adaptively shifts among binary, multiple-choice, and open-ended question formats to reduce burden. The paper evaluates this framework on webpage generation and banner generation, reporting improved alignment-cost tradeoffs relative to several direct-questioning baselines, including fixed-format, flexible-format, and retrieval-augmented variants.

**Compliance With Llm Reviewing Policy:**

Affirmed.

**Final Justification:**

The rebuttal addressed my main concerns.

**Key Questions For Authors:**

1. How well does the proxy burden correlate with real human effort in this exact task setting? The paper cites external evidence, but does not validate the proxy on the design benchmarks here. Even a small-scale human study measuring response time, perceived effort, or completion fatigue would significantly strengthen the paper.

2. How much of the gain comes from adaptive format selection versus the MPC-style replanning itself? The ablations are useful, but I would like a cleaner decomposition. For example, what happens with a strong heuristic format scheduler without question plans or respond-or-reject control? This would clarify whether the main benefit is the control formulation or simply format adaptation.

3. How sensitive are the results to model choice and cross-model evaluation? Since the setup uses LLMs for user simulation, plan updating, generation, and judging, it would help to show stronger main-paper evidence that the gains are not tied to one model family.

4. Why is the generation performed only once after clarification, rather than allowing interleaved generation and clarification? I understand the desire to isolate questioning policy, but this also makes the setting somewhat stylized. A discussion of how A-MPQC would extend to iterative generation-refinement loops would improve the paper’s practical relevance.

**Limitations:**

yes

**Strengths And Weaknesses:**

Strengths:

- This paper has several clear strengths. First, the problem formulation is meaningful and timely: instead of optimizing only final output quality, it explicitly treats user burden as a resource, which is a useful perspective for human-agent interaction in design settings.

- Second, the paper proposes a coherent method rather than a loose heuristic collection. The three components fit together conceptually and are clearly motivated by the difficulty of applying standard MPC directly when user interactions themselves are costly.

- Third, the experimental section is reasonably structured: the paper compares against multiple baselines under a matched budget, studies budget allocation, and includes an ablation of the main components. The reported results suggest that A-MPQC achieves a better cost-quality tradeoff than the compared baselines on both tasks.


Weaknesses:
- The main weakness is soundness of evaluation. The paper’s central claim is about reducing human cognitive burden, yet both the user and the burden signal are simulated through LLM proxies. In particular, the cost metric is defined as the product of model-side reasoning tokens and response tokens, and the justification is largely indirect. This may be a reasonable first proxy, but it is still a strong assumption, and the current evidence does not establish that the reported gains would transfer to real users. The paper itself acknowledges that the LLM user proxy is imperfect and should not substitute for real-user studies, which is appropriate, but this limitation substantially weakens the strength of the empirical claims.

- A second weakness is that the empirical pipeline stacks several learned components: frozen LLMs for plan updating, simulated user responses, and LLM-based judging of similarity. This makes the evaluation vulnerable to hidden coupling effects. For example, on WebGen-V the main paper uses Gemini-2.5-Pro for the user agent, question agent, generation, and judging, which raises concern that some of the measured gains may partly reflect model-specific coordination rather than a generally stronger interaction policy.

- A third weakness is that originality is mixed. The paper is not merely repackaging an existing approach, because the cost-aware clarification formulation and the specific interaction-level MPC reinterpretation are interesting. However, the novelty is primarily in problem framing and test-time orchestration rather than in a new learning algorithm or a deeply developed theory. That is still acceptable, but it means the paper’s impact depends heavily on the credibility of its evaluation, which is currently limited by simulation-only evidence.

---

> ### Author Rebuttal · Authors · 2026-03-30
>
> We thank the reviewer for the thoughtful and constructive feedback. We address each concern point-by-point below.
>
> > [W1, Q1] How well does the proxy burden correlate with real human effort in this exact task setting?
>
> We agree that task-specific validation is important. We therefore conducted a small-scale human validation study on the same multi-turn visual-design interactions and observed positive Spearman/Kendall rank correlations between human-rated burden and our token-based proxy, suggesting that the proxy tracks the relative burden trends in this exact setting. Please see our response to Reviewer t7e5 (W3/Q2) for the detailed protocol and statistics.
>
> > [Q2] How much of the gain comes from adaptive format selection versus the MPC-style replanning itself? For example, what happens with a strong heuristic format scheduler without question plans or respond-or-reject control?
>
> Our Table 2 already provides most of this decomposition. In particular, the ablation w/o Design I (no question plans) isolates the effect of removing lookahead planning, and the cleaner ablation discussed in our response to Reviewer mAtH (W4) shows that removing the use of rejection signals for replanning also degrades performance. Together, these results indicate that the gain is not reducible to format adaptation alone.
> Importantly, DG + Flexible already serves as a strong heuristic format scheduler: it can choose among binary, multiple-choice, and open-ended formats, but it does not use question plans or respond-or-reject control. A-MPQC can be viewed as building on top of this flexible-format baseline by adding our three proposed 3 designs: interaction-level MPC view, question plans, and respond-or-reject control.
>
> > [W2, Q3] How sensitive are the results to model choice and cross-model evaluation?
>
> We already include a cross-model result in Appendix Table 3, where the question agent is replaced with GPT-5, and the overall conclusion remains unchanged. This suggests that the gain is not tied to a specific question-agent backbone. Our evaluation also follows the original benchmark protocol (Sec. 5.1.3), and the judge is used to assess similarity to the target design rather than assign an unconstrained standalone quality score, which helps reduce concern about generic LLM-as-a-judge bias.
>
> To further test robustness, we additionally ran GPT-5 as both the user agent and the question agent on MIMO. The qualitative conclusion remains the same: A-MPQC stays in the low-cost / strong-improvement region. Together with the additional budget results in response to Reviewer Afw1 (Q2/Q4), this suggests that the observed tradeoff is not tied to one specific budget split.
>
> | Method| C | $\Delta S$ | $\Delta S/C$ |
> | --- | ---: | ---: | ---: |
> | DG| 6.55 | 0.41 |0.06 |
> | DG + Binary | 0.05 | 0.00 | 0.00 |
> | DG + Multiple-Choice |1.36 |0.08 | 0.06 |
> | DG + Open-Ended | 6.79 | 0.18 | 0.03 |
> | DG + Flexible|3.17 | 0.46 | 0.15 |
> | DG + Flexible + RAG| 2.39 |0.35 | 0.15 |
> | **A-MPQC** |0.23 | 0.39 | **1.69** |
>
> > [W3] Originality is mixed.
>
> Our novelty instead lies in introducing a new problem formulation and a corresponding test-time control framework: to our knowledge, prior work has not formulated multi-round clarification itself as a trajectory-optimization problem with user burden as an explicit constrained resource. In addition, prior MPC-style approaches typically rely on learned dynamics or reward models, whereas A-MPQC makes replanning feasible in costly human interaction through question plans, respond-or-reject surrogate rewards, and adaptive format control at test time. We therefore see the contribution not as prompt orchestration alone, but as a concrete interaction-level control formulation supported by component-wise empirical evidence. We agree that stronger human validation would further strengthen the paper, and we added preliminary human-subject evidence to address this concern.
>
> > [Q4] Why is the generation performed only once after clarification, rather than allowing interleaved generation and clarification?
>
> We intentionally decouple intent elicitation from downstream refinement in Sec. 3 (Problem Setting) to isolate the effect of the questioning policy itself. If generation is interleaved after every turn, gains from better questioning and gains from iterative artifact refinement become confounded. There is also a practical reason: text interaction is relatively cheap, while repeatedly rendering webpages or banners can be expensive and slow for users. That said, our framework is compatible with iterative refinement loops. We view the current paper as isolating the “clarify before generate” component, and extending A-MPQC to alternating generate/clarify cycles is a natural future work.

---

> > ### Author Rebuttal · Reviewer_kbxB · 2026-04-04
> >
> > Thanks for the response,  I've increased my score.

---

> > > ### Author Response · Authors · 2026-04-06
> > >
> > > We sincerely thank the reviewer for the thoughtful follow-up and for increasing the score. We are greatly encouraged by the positive assessment in the written comment.
> > >
> > > At the same time, we noticed that the acknowledgement was marked as **partially resolved**, although the comment itself does not seem to indicate any specific remaining follow-up question. If there are still any points that the reviewer would like us to further clarify, we would be truly grateful for the opportunity to address them.
> > >
> > > Thank you again for your constructive feedback. Your comments have helped us improve the paper significantly.

---

### Official Review · Reviewer_Afw1 · 2026-03-12

**Soundness:** 3
**Presentation:** 4
**Significance:** 3
**Originality:** 3
**Overall Recommendation:** 4
**Confidence:** 4

**Summary:**

The manuscript studies a pressing issue in human–agent interaction for visual design, where users often struggle to express their design intent through a single prompt and must rely on multiple rounds of clarification. The paper investigates how an intelligent agent can ask clarification questions that both improve alignment between generated designs and user intent and reduce the cognitive burden on the user. To address this problem, the authors propose Agentic Model Predictive Questioning Control (A-MPQC), a test-time framework that formulates multi-round clarification as a trajectory optimization problem under a fixed question budget. The framework models the interaction process as iterative plan updates, where the agent asks questions, receives responses, updates a design plan, and finally generates the visual artifact from the resulting plan.
The research's principal concept is to apply ideas from Model Predictive Control to guide the questioning policy during human-agent interaction without additional model training. The proposed method introduces question plans with short-horizon lookahead, a respond-or-reject surrogate reward that signals whether a questioning direction is useful, and adaptive question formats that move from expressive questions to lower-burden formats when possible. Experiments on webpage generation and banner image generation benchmarks demonstrate that the approach improves alignment with user intent while reducing interaction cost compared with several baseline questioning strategies. The results suggest that optimizing questioning policies at test time can produce a more favorable trade-off between output quality and human cognitive effort in interactive visual design systems.

**Compliance With Llm Reviewing Policy:**

Affirmed.

**Final Justification:**

Thanks for the response, I've increased my score.

**Key Questions For Authors:**

1. The experiments rely on simulated users implemented with LLMs to approximate human responses and cognitive cost. Could the authors provide evidence that this proxy reliably reflects real human interaction behavior? For example, have the authors conducted or considered small-scale human studies to validate the interaction cost proxy? A convincing validation could strengthen the empirical soundness of the work.
2. The paper models user interaction cost using token-based proxies (reasoning tokens and response tokens). Could the authors clarify how robust this proxy is across different models and tasks? Additional analysis or justification would help determine whether the reported cost improvements reflect meaningful reductions in human effort.
3. The method is evaluated on webpage generation and banner generation. How well would A-MPQC generalize to other interactive generation settings, such as text generation, multimodal reasoning, or general VQA tasks? Evidence or discussion of broader applicability could increase the significance of the work.
4. The framework depends on several interaction parameters(e.g., question budget, number of rounds,and per-round questions) Could the authors provide further analysis on how sensitive the method is to these choices? Understanding this would clarify whether the approach is robust in practical deployments.

**Limitations:**

yes

**Strengths And Weaknesses:**

The paper is technically sound and presents a well-motivated framework for improving human–agent interaction in visual design tasks. The formulation of multi-round clarification as a trajectory optimization problem is methodologically reasonable, and the use of a test-time control policy avoids the need for additional training data or model retraining. The proposed A-MPQC framework integrates several coherent components, including question planning with short-horizon lookahead, a respond-or-reject surrogate reward, and adaptive question formats, which together form a structured questioning policy. The empirical evaluation on two visual design benchmarks (webpage generation and banner generation) is generally appropriate and compares against several baselines under a controlled interaction budget. The results indicate improved alignment–cost tradeoffs, suggesting the approach is practical for interactive generation settings where user effort must be minimized. The paper is also generally well structured and clearly explains the interaction protocol, cost formulation, and algorithmic procedure.
Despite the clear framework, several aspects limit the strength of the claims. First, the evaluation relies on simulated users implemented with large language models rather than real human participants, which may not accurately capture human cognitive burden or interaction behavior. Second, some design choices, such as the token-based proxy for cognitive cost and the binary respond-or-reject reward, are heuristic and lack deeper theoretical justification. Third, the method largely combines existing concepts from model predictive control, interactive clarification, and test-time optimization, so the level of methodological novelty may be moderate. Finally, the experiments focus on two specific visual design tasks, and it remains unclear how well the approach generalizes to other multimodal generation settings or real-world interactive systems.

---

> ### Author Rebuttal · Authors · 2026-03-30
>
> We thank the reviewer for the thoughtful and constructive feedback. We address each concern point-by-point below.
>
>
> > [W1, Q1] Could the authors provide evidence that this proxy reliably reflects real human interaction behavior?
>
> We agree this is a central question for the paper. To address it, we supplemented the simulation results with a small-scale human validation study on the same multi-turn interactions; the results show positive rank correlation between human-rated burden and our token-based proxy, providing preliminary evidence that it captures meaningful aspects of perceived burden in this setting. Please see our response to Reviewer t7e5 (W3/Q2) for the detailed protocol and full results.
>
> > [W2] Some design choices are heuristic.
>
> The proxy is intentionally lightweight because a richer reward would itself require more user effort. The binary respond-or-reject signal in Sec. 4.2.2 is therefore not just a UI choice; it is the lowest-cost control signal that still tells the agent whether a questioning direction is worth pursuing. Importantly, rejection is not free: it still incurs reasoning and response cost.
>
> > [W3] The method largely combines existing concepts, so the level of methodological novelty may be moderate.
>
> The novelty lies in both the problem formulation and the interaction-level control design. To our knowledge, prior clarification work has not formulated multi-round clarification itself as a trajectory-optimization problem, and most existing methods mainly optimize alignment or information gain without explicitly treating user burden as a constrained resource. At the same time, prior MPC-based approaches typically rely on learned dynamics or reward models, whereas A-MPQC makes replanning feasible in costly human interaction through question plans, respond-or-reject surrogate rewards, and adaptive format control at test time. The final empirical results further show that these design choices are effective, which we believe constitutes the core novelty of A-MPQC.
>
>
> > [W4, Q3] How well the approach generalizes to other multimodal generation settings or real-world interactive systems?
>
> We view A-MPQC as most natural for interactive generation problems with three properties: partially observed user intent, an explicit intermediate plan/state that can be updated through clarification, and a final artifact generated from that clarified plan. Webpage and banner generation fit this especially well, which is why we chose them as first testbeds. In contrast, tasks such as general VQA or multimodal reasoning do not need to expose such an intermediate plan, so broader transfer is best framed as a promising direction.
>
> > [Q2,Q4] Could the authors clarify how robust this proxy is across different models ? Could the authors provide further analysis on how sensitive the method is to several interaction parameters?
>
> We chose (B=12) in the main paper following [1], who evaluate with 15-turn conversations and report that key alignment signals often plateau after around 10 interactions; we therefore use (B=12) as a conservative middle ground. Fig. 4 already studies sensitivity to budget allocation under this fixed budget by varying ((n,m)) across ((12,1), (6,2), (3,4), (2,6)), and ((1,12)). Beyond this allocation study at (B=12), we additionally ran a smaller-budget setting using GPT-5 as both the question agent and the user agent. This serves two purposes: it tests whether the method remains stable under a different interaction budget, and whether the cost–performance tradeoff remains consistent under a different model family rather than depending on Gemini-specific behavior. The same qualitative pattern still holds: A-MPQC remains in the low-cost / strong-improvement region relative to the baselines. This suggests that the method is not fragile to the specific choice of (B=12), and that the usefulness of the token-based proxy is not limited to one particular model setup. While token scales differ across models, the comparative conclusion remains stable.
>
> ### 3 rounds × 4 questions (GPT-5 as question agent and user agent)
>
> | Method| C | $\Delta S$ | $\Delta S/C$ |
> | --- | ---: | ---: | ---: |
> | DG| 6.55 | 0.41 |0.06 |
> | DG + Binary | 0.05 | 0.00 | 0.00 |
> | DG + Multiple-Choice |1.36 |0.08 | 0.06 |
> | DG + Open-Ended | 6.79 | 0.18 | 0.03 |
> | DG + Flexible|3.17 | 0.46 | 0.15 |
> | DG + Flexible + RAG| 2.39 |0.35 | 0.15 |
> | **A-MPQC** |0.23 | 0.39 | **1.69** |
>
> ### 2 rounds × 3 questions (GPT-5 as question agent and user agent)
>
> | Method| C | $\Delta S$ | $\Delta S/C$ |
> | ---- | ---: | ---: | ---: |
> | DG + Binary | 0.03 | 0.04 | 1.33 |
> | DG + Multiple-Choice | 0.01 | 0.03 | 3.00 |
> | DG + Open-Ended | 1.12 | 0.26 | 0.23 |
> | DG + Flexible | 1.32 | 0.41 | 0.31 |
> | DG | 1.45 | 0.37 | 0.26 |
> | DG + Flexible + RAG | 0.96 | 0.33 | 0.34 |
> | **A-MPQC** | 0.10 | 0.36 | **3.60** |
>
> [1] Proactive agents for multi-turn textto-image generation under uncertainty, ICML 2025

---

> > ### Author Rebuttal · Reviewer_Afw1 · 2026-04-05
> >
> > Thanks for the response, I've increased my score.

---

> > > ### Author Response · Authors · 2026-04-06
> > >
> > > We sincerely thank the reviewer for the thoughtful follow-up and, in particular, for explicitly stating that the concerns have been **fully resolved**.
> > >
> > > We are greatly encouraged by this very positive assessment. Since the reviewer now finds that the key issues have been adequately addressed, and considering that this work is, to the best of our knowledge, the first to explicitly study and optimize the trade-off between **generation accuracy** and **human burden**, we would be very grateful if the overall score could be reconsidered so as to better reflect this updated evaluation.
> > >
> > > Thank you again for your insightful and constructive comments. Your feedback has genuinely helped us improve the paper, and we are truly grateful for it!

---

### Official Review · Reviewer_mAtH · 2026-03-13

**Soundness:** 3
**Presentation:** 3
**Significance:** 3
**Originality:** 3
**Overall Recommendation:** 4
**Confidence:** 2

**Summary:**

This paper proposes A-MPQC, a test-time framework for interactive visual design agents that frames clarification as a receding-horizon MPC-style process. At each step the agent generates a question plan, asks a single question, and updates the plan based on user feedback, with the goal of reducing user burden under a fixed question budget. Experiments on WebGen-V and MIMO show efficiency gains over direct generation baselines.

**Compliance With Llm Reviewing Policy:**

Affirmed.

**Final Justification:**

The paper targets a practical problem: reducing user-side cost in interactive design agents under a fixed question budget. The method requires no retraining, and experiments on WebGen-V and MIMO show efficiency gains over direct generation. The rebuttal resolved two of my four original concerns: the protocol-matched refusal experiment (W2) ruled out a mechanical artifact, and the WebGen-V ablation (W4) generalized the component analysis.

Two concerns remain partially addressed. On the burden proxy (W1), the authors argue Spearman rho=0.41 is "moderate" and the 6 raters are domain-relevant researchers. The correlation is not strong, and the raters are not representative end users. The claim about reducing "human burden" should be qualified in a revision. On the MPC framing (W3), the follow-up clarifies the intended connection at the state/action/reward level and acknowledges the difference from simulator-based MPC. This is reasonable, revising Sec. 4.2 and Fig. 3 as proposed would address it.


Therefore, I would like to keep my recommendation as 4.

**Key Questions For Authors:**

See weakness above

**Strengths And Weaknesses:**

**Strengths**
- The problem is well-motivated: most prior interactive agent work ignores user-side cost, and the explicit cost-quality framing makes this concrete.
- The method requires no retraining and the state/action structure (plan, question, plan update) is straightforward to follow.
- Experiments on WebGen-V and MIMO show efficiency gains over direct generation baselines.

**Weaknesses**
- My main concern is about whether the burden proxy is meaningful. The cost C is computed from reasoning tokens T^think from Gemini-2.5-Pro, which is from one model's API rather than a general measure of human cognitive effort. It is not clear to me whether this correlates with actual human response time or effort for visual design questions, and the claim of "reduced human burden" seems hard to evaluate without either a human study or at minimum a sensitivity check across different user simulators.
- From my understanding, A-MPQC enables a respond-or-reject option for the user, while all baselines use answer-only, meaning a rejection costs only 2 tokens while baselines must produce a full response. I am wondering if baselines were given the same refusal channel with the same cost accounting, would the efficiency gap narrow substantially?
- It is not clear to me that "MPC" describes something beyond structured replanning via prompting. Algorithm 1 generates a single question plan per step and uses accept/reject as a surrogate signal, but there is no search over candidate trajectories. I would like to learn from the authors what is being optimized beyond LLM-based replanning, and whether they tried K>1 candidate rollouts at each step to justify the MPC framing.
- The component analysis in Table 2 is only on MIMO, which makes it hard to know whether the findings generalize to WebGen-V where the ambiguity and cost structure differ. Also, removing "Design II" switches the user to always-respond, which changes the environment rather than just the agent's decision rule. Cleaner ablations that hold the user policy fixed would strengthen this section.

---

> ### Author Rebuttal · Authors · 2026-03-30
>
> We thank the reviewer for the thoughtful and constructive feedback. We address each concern point-by-point below.
>
> > [W1] My main concern is about whether the burden proxy is meaningful.
>
> Please see our response to Reviewer t7e5 (W3/Q2). Briefly, we conducted a small-scale human validation study on the same multi-turn interactions and found positive rank correlations between human-rated burden and our token-based proxy, providing preliminary support that the proxy captures meaningful aspects of perceived interaction burden.
>
> > [W2] If baselines were given the same refusal channel, would the efficiency gap narrow substantially?
>
> We reran the MIMO comparison by giving the baselines the same refusal channel and the same cost accounting. The gap becomes smaller, but it does not disappear: refusal alone helps a strong baseline (e.g., DG, DG + Flexible), but A-MPQC remains best overall.
> This supports our central claim that the gain is not merely from allowing a cheap “reject,” but from using that signal to replan both questioning direction and format.
>
> ### MIMO with refusal-enabled baselines
>
> | Method  | Refusal Channel | C | $\Delta S$ | $\Delta S/C$ |
> | --- | --- | ---: | ---: | ---: |
> | DG | No | 6.65 | 0.26 | 0.04 |
> | DG + Binary | No | 0.06 | 0.02 | 0.34 |
> | DG + Multiple-Choice | No |1.61 | 0.07 |0.04 |
> | DG + Open-Ended | No| 5.24 | 0.21 | 0.04 |
> | DG + Flexible| No| 2.32 | 0.46 |0.20 |
> | DG + refusal| Yes|1.86 | 0.31 |0.17 |
> | DG + Binary + refusal | Yes|0.32 |0.01 |0.03 |
> | DG + Multiple-Choice + refusal | Yes|0.31 |-0.02 |- |
> | DG + Open-Ended + refusal | Yes |2.80 |0.04 |0.01 |
> | DG + Flexible + refusal | Yes|1.41 |0.41 |0.29 |
> | **A-MPQC**| Yes|1.24 |  0.59 | **0.48** |
>
> Note. “-” means the measured similarity improvement was negative, so the efficiency ratio is not meaningful.
>
> > [W3] Algorithm 1 generates a single question plan per step and uses accept/reject as a surrogate signal. Whether they tried K>1 candidate rollouts at each step to justify the MPC framing.
>
> A-MPQC is not merely single-path LLM-based replanning. We formulate the whole multi-round clarification process, with (n) rounds and (m) questions per round, as a receding-horizon control problem, where the horizon is the (n) interaction rounds. Within each round, Algorithm 1 generates and executes multiple question plans through the for-loop, corresponding to (K=m). Thus, the method is not a one-shot prompt rewrite; under a fixed interaction budget, it repeatedly updates the design plan and questioning policy using surrogate rewards to optimize the alignment–burden tradeoff.
>
> We understand that the reviewer’s notion of (K>1) rollout is closer to standard MPC, where one first collects real user feedback for multiple candidate trajectories and then selects one to continue. This is indeed a more direct MPC implementation, and we already discuss it in Sec. 4.3. This would require extra user feedback for all rollout candidates, so human burden grows with (K), which conflicts with our cost-aware objective. Table 2 supports this directly: under the more literal MPC-style control in A-MPQC-on-m with (K=3), the similarity is close to A-MPQC, but the cost rises sharply from 1.238 to 6.696, causing efficiency to drop substantially. Our method is therefore better understood as interaction-level MPC, rather than explicit trajectory search over all candidate futures.
>
>
> > [W4] The component analysis in Table 2 is only on MIMO, which makes it hard to know whether the findings generalize to WebGen-V. Cleaner ablations that hold the user policy fixed would strengthen this section.
>
> We agree and have now run additional WebGen-V ablations, including a cleaner variant that keeps the user policy fixed and removes only the agent’s use of rejection for replanning. The trend is consistent with MIMO: full A-MPQC remains the strongest overall variant, and every key design contributes. The cleaner ablation also clarifies that the gain is not simply from letting the user reject, but from exploiting rejection as a control signal.
>
> ###  Ablation Study on WebGen-V
>
> | Variant  | C | $\Delta S$ | $\Delta S/C$ |
> | --- | ---: | ---: | ---: |
> | w/o Design I (no question plans)|3.56 |0.30 |0.08 |
> | w/o Design II (all-respond)|2.30 |0.28 |0.12 |
> | A-MPQC-IgnoreReject|2.28 |0.13 |0.06 |
> | w/o Design III (direct questioning) |2.64 |0.28 |0.10 |
> | **A-MPQC (full)**|2.55 |0.48 | **0.19** |
>
>
> ### Cleaner ablation for Design II on MIMO
>
> | Variant| User policy | Uses reject for replanning? | C | $\Delta S$ | $\Delta S/C$ |
> | --- | --- | --- | ---: | ---: | ---: |
> | w/o Design II (all-respond) | always-respond| No| 0.47 | 0.20 | 0.43 |
> | A-MPQC-IgnoreReject| respond-or-reject | No|0.86 | 0.31 | 0.36|
> | **A-MPQC (full)**| respond-or-reject | Yes|1.24 |0.59 | **0.48** |
>
> We will revise Sec. 5.4 to make this distinction explicit.

---

> > ### Author Rebuttal · Reviewer_mAtH · 2026-04-04
> >
> > Thank you for the response. W2 (refusal channel) and W4 (ablation generalization) are addressed.
> >
> > **W1 (burden proxy).** The human study helps but the reasoning burden correlation is weak (rho=0.41, tau=0.28). Reasoning tokens are a substantial component of the proxy C. The study also uses 6 LLM experts, not the target users (designers, end users). The claim about reducing "human burden" should be qualified.
> >
> > **W3 (MPC framing).** The authors argue the for-loop over m questions constitutes K=m candidate rollouts. But these questions are executed sequentially with real user feedback between them. In standard MPC, candidate rollouts are evaluated before committing to an action. The A-MPQC-on-m variant (K=3, cost from 1.24 to 6.70) confirms the method cannot do what the MPC framing implies. Reviewer t7e5 raises the same concern (W2). The system is structured replanning with surrogate feedback, and the paper should say that.
> >
> > Therefore, I would like to keep my overall recommendation at 4 given W1 and W3.

---

> > > ### Author Response · Authors · 2026-04-06
> > >
> > > We sincerely thank the reviewer for the thoughtful follow-up and for the positive overall assessment. We are also grateful that W2 and W4 were resolved by the additional experiments. We address the remaining concerns below.
> > >
> > > > **W1 (burden proxy)**
> > >
> > > We agree that the human study should be interpreted cautiously and that the current evidence should not be overstated. However, we would like to clarify two points:
> > >
> > > First, the reasoning-burden correlation is not strong, but we believe it is more accurately described as moderate under Spearman’s ρ = 0.41 and weak-to-moderate under Kendall’s τ = 0.28, given the ranges of both are (-1, 1).
> > >
> > > Second, the six raters were not generic LLM experts; they are researchers with direct experience in agentic banner generation and HTML generation, and are therefore domain-relevant expert evaluators for design-generation tasks. While they are not a complete substitute for professional designers or end users, we believe they are appropriate expert users for this study.
> > >
> > > We will revise the wording in the paper to qualify the claim accordingly.
> > >
> > > > **W3 (MPC framing)**
> > >
> > > We thank the reviewer again for the helpful follow-up. We would like to clarify more precisely what correspondence to MPC we intend, and where our setting differs from the common simulator-based formulation.
> > >
> > > We do agree with you that our setting differs from the most common MPC setup, where candidate trajectories are evaluated cheaply using a simulator or a learned dynamics/reward model before executing the first action. However, in our multi-round clarification setting, candidate evaluation itself requires real user feedback and therefore incurs user burden. This is exactly why we introduce A-MPQC: not to abandon the MPC formulation, but to adapt it to a costly human-in-the-loop setting.
> > >
> > > The connection to MPC is at the level of the **state / action / reward / transition framing** under receding-horizon control. The **state** is the current design plan, the **action** is a proposed questioning direction, and the user response is parsed into a **surrogate reward signal** for that direction. Under this view, **asking a question is not yet committing to an action** in the plan space. Instead, the question is first evaluated through user feedback, and only **accepted** questions provide the substantive QA information that is incorporated into the design plan and moves the system to the next state. In this sense, A-MPQC still evaluates a questioning direction before committing to it, but the evaluation is performed through costly user interaction rather than through a free simulator or learned reward model. This distinction is also a key source of A-MPQC’s novelty.
> > >
> > > Our goal is therefore not to claim an implementation identical to simulator-based MPC, but to emphasize that the MPC perspective remains meaningful in A-MPQC even when the environment is a real user and feedback itself is expensive. To make this intended correspondence clearer, we will revise Sec. 4.2 and Fig. 3 to emphasize both the alignment with the MPC view and the specific difference from standard MPC in settings where rollout evaluation is effectively free.
> > >
> > > ---
> > >
> > > In light of these clarifications and the additional experiments we added in response to the reviewer’s suggestions, we would sincerely appreciate it if the reviewer could kindly take them into account in the final assessment.

---

### Decision · Program_Chairs · 2026-04-30

**Decision:**

Accept (regular)

**Comment:**

This paper proposes A-MPQC, a test-time framework for multi-round clarification in visual design that explicitly optimizes the tradeoff between final design alignment and user interaction burden under a fixed question budget. Reviewers were broadly aligned that this is a timely and practically relevant problem, that the method is coherent and easy to follow, and that the empirical results are strong for the scope of the paper.

In the main experiments, A-MPQC achieves the best efficiency among compared methods on both WebGen-V and MIMO, while the ablations suggest that question plans, reject/respond replanning, and adaptive question formats each contribute to the gains. The discussion record also indicates a uniformly positive final panel, with the authors reporting that the three initially borderline reviewers raised their scores to 4, leaving all four reviews positive.

The concerns that mattered most for the decision were all about soundness and claim calibration, not about whether the paper had a real signal. The central issues were: whether the token-based burden proxy and LLM-user simulation are sufficient to support claims about “human burden”; whether the MPC language overstates what is closer to structured replanning with surrogate feedback; whether the respond/reject channel gives A-MPQC an unfair mechanical advantage; whether the ablations were clean enough and generalized beyond MIMO; and whether the conclusions were too dependent on a particular model stack or benchmark setup. The rebuttal substantially improved the case for the paper by adding a small human validation study, refusal-enabled baseline comparisons, cleaner WebGen-V ablations, and additional cross-model results. Remaining limitations are real, but they now look like camera-ready revisions rather than reasons to reject.

Reviewer mAtH: score stayed at 4. The rebuttal resolved two of four concerns, but not enough to move the reviewer above weak accept. All other reviewers raised the score to 4.